# An emergency response model for the formation and dispersion of plumes originating from major fires (BUOYANT v4.20)

Jaakko Kukkonen[1,2], Juha Nikmo[1], Kari Riikonen[1], Ilmo Westerholm[1], Pekko Ilvessalo[1], Tuomo Bergman[1] and Klaus Haikarainen[1]

[1]Finnish Meteorological Institute, Erik Palménin aukio 1, P.O. Box 503, 00101, Helsinki, Finland
[2]Centre for Atmospheric and Climate Physics Research, and Centre for Climate Change Research, University of Hertfordshire; College Lane, Hatfield, AL10 9AB, UK

*Correspondence to*: Jaakko Kukkonen (jaakko.kukkonen@fmi.fi)

**Abstract.** A mathematical model called BUOYANT has previously been developed for the evaluation of the dispersion of positively buoyant plumes originating from major warehouse fires. The model addresses the variations of the cross-plume integrated properties (i.e., the average properties along a trajectory) of a rising plume in a vertically varying atmosphere, and the atmospheric dispersion after the plume rise regime. We have described in this article an extension of the BUOYANT model to include a detailed treatment of the early evolution of the fire plumes, before the plume rise and atmospheric dispersion regimes. The input and output of the new module consist of selected characteristics of forest or pool fires, and the properties of a source term for the plume rise module, respectively. The main structure of this source term module is based on the differential equations for low-momentum releases of buoyant material, which govern the evolution of the plume radius, velocity and density differences. The source term module is also partially based on various experimental results on fire plumes. We have evaluated the refined BUOYANT model by comparing the model predictions against the experimental field-scale data of the Prescribed Fire Combustion and Atmospheric Dynamics Research Experiment, RxCADRE. The predicted concentrations of $CO_2$ agreed fairly well with the aircraft measurements conducted in the RxCADRE campaign. We have also compiled an operational version of the model. The operational model can be used for emergency contingency planning and the training of emergency personnel, in case of major forest and pool fires.

## 1   Introduction

The dispersing of fire plumes can represent a substantial hazard to the health of people and the state of the environment, in addition to the direct effects of major fires at the accident sites. Major fires include,

e.g., fires in warehouses and industrial sites, and wildland fires. The latter category includes, e.g., heath, moorland and forest fires. Major wildland fires can result in substantially more extensive and intensive fire plumes, compared with industrial and warehouse fires. The emissions from wildland fires can affect human health, the state of ecosystems, carbon stocks, and land surface reflectance.

Solid fuels, such as vegetative biomass, are consumed in a two-stage process including pyrolysis and combustion (Ottmar, 2014). Pyrolysis is the chemical decomposition of a solid material by heating, and it results in gaseous and solid products (Ottmar, 2014; Rein, 2016). Pyrolysis is an endothermic process, and by definition, it does not involve combustion reactions. Although both stages occur nearly

simultaneously, pyrolysis occurs first, followed by oxidation of the escaping hydrocarbon vapours (Ottmar, 2014; Rein, 2016).

Combustible liquids are during burning commonly first converted from liquid to vapour, without any chemical decomposition of fuel molecules (Drysdale, 2016). Exceptions to this general rule include high

molecular weight liquids, which may be subject to chemical decomposition at temperatures associated with vapour formation (Drysdale, 2016). In the atmosphere, the air is mixed and reacts with the flammable vapours that are evaporated from the liquid surface; this is called burning as a diffusion flame. Complete combustion is rarely achieved in atmospheric conditions; commonly products of incomplete combustion will therefore be formed, such as carbon monoxide and smoke (Khan et al., 2016). Hundreds

of chemical compounds are emitted into the atmosphere during wildland fires (Ottmar, 2014).

In realistic simulations of air quality and climate, one needs to include wild-land fire emission models or inventories (e.g., Saarnio et al., 2010; Sofiev et al., 2009; Wiedinmyer et al., 2006). The available estimation methods on a global scale have been addressed by, e.g., Hoelzemann et al. (2004), Ito and

Penner (2004) and van der Werf et al. (2003). These inventories include fires at varying horizontal resolutions, from 1 km to 1° (latitude and longitude), and typically use a monthly temporal resolution. Wild-land fire emission predictions have been made also for specific episodes, fires, and regions (e.g., Dennis et al., 2002; Lavoué et al., 2000; Soja et al., 2004; Anderson et al., 2004).

Both inventory-type and satellite-based emission estimates of wildland fires may use as input values for the burned area, fuel loads, and combustion completeness to calculate emissions (Seiler and Crutzen,

1980). However, studies using satellite-measured fire radiative power (FRP) do not require these input datasets. Such methods directly integrate FRP to the total fire radiative energy (FRE), which is in turn related to total emissions (Sofiev et al., 2009, van der Werf et al., 2010). Such modelling methods are valuable, especially for online inventories of wildfires and their dispersion on regional, continental and global scales (e.g., https://silam.fmi.fi/).

The satellite-based methods can be considered to be complementary in scope, compared to the detailed emission, plume rise and dispersion modelling presented in the current article. The main advantages of the satellite-based methods are that these can be used online and for extensive areas containing wild-land fires. Limitations of the satellite-based methods include that these cannot detect smaller fires; these also do not contain any direct information on the nature of the fire (e.g., forest fire or industrial fire). If more detailed and accurate analyses for any specific fire will be needed, then it is advisable to use a dedicated fire emission and plume rise model.

Some studies have collected data sets that include fuel characteristics and consumption, and environmental variables, regarding both wildfires and prescribed fires (Ottmar, 2014). These datasets have been used for fuel consumption models, such as Consume (Prichard et al., 2007), FOFEM (Reinhardt et al., 1997), and BORFIRE (de Groot et al., 2007 and 2009).

Semi-empirical and computational fluid dynamics (CFD) models have been used for the modelling of liquid pool fires (Rew and Hulbert, 1996; Brambilla and Manca, 2009). The CFD models have not yet been applied for the dispersion of wildland fires. Semi-empirical models use heat transfer principles and dimensional analysis to predict, e.g., the fuel burning rate, the fraction of radiative power and the flame length. The CFD models for liquid pool fires solve the Navier-Stokes equations of fluid flow, together with treatments for the chemical and physical processes occurring in fires. There are numerous studies on hydrocarbon pool fires, including both experimental data (Mudan, 1984; Koseki, 1989; Luketa-Hanlin, 2006; Raj, 2007b; Fingas, 2014; Beyler, 2016) and theoretical analyses (Hottel, 1959; Hertzberg, 1973; Babrauskas, 1983; Hostikka et al., 2003; Fay, 2006; Raj, 2007a).

The pollutants from wildland fires are commonly transported in the troposphere but may in some cases reach the stratosphere (e.g., Freitas et al., 2007; Sofiev et al., 2012). The most important process in

describing the plume dispersion from major fires can be considered to be the initial plume rise (e.g., Liousse et al., 1996; Trentmann et al., 2002; Kukkonen et al., 2014). Overviews of buoyant plume models have been presented, for example, by Devenish et al. (2010), Jirka (2004), Lareau and Clements (2017), Paugam et al. (2016) and Val Martin et al. (2012).

The modelling in this study is partly based on previous work by Ramsdale et al. (1997) and Kukkonen et al. (2000). The mathematical description of the modelling of plume rise and near-field dispersion was reported in Martin et al. (1997). A module for the larger-scale dispersion was added to the modelling system, as described by Nikmo et al. (1997 and 1999). A mathematical model called BUOYANT has been developed, partly based on these previous studies; a previous version of this model has been described in detail by Kukkonen et al. (2014). The model is applicable for predicting the initial plume rise and atmospheric dispersion of pollutants originating from strongly buoyant sources. The model includes treatments for the cross-plume integrated properties of a buoyant plume in a vertically varying atmosphere. In particular, the model allows for a rising buoyant plume that interacts with an inversion layer.

The plume rise predictions of the BUOYANT model have previously been evaluated against two experimental field data sets regarding prescribed wild-land fires (Kukkonen et al., 2014). These were the "Smoke, Clouds and Radiation – California" experiment (SCAR-C) in Quinault in the US in 1994 (e.g., Kaufman et al., 1996; Gassó and Hegg, 1998) and an experiment in Hyytiälä in Finland in 2009 (e.g., Virkkula et al., 2014a and 2014b). E.g., for the SCAR-C experiment, the modelled vertical ranges of the plume at maximum plume rises were from 500 to 800 m and from 200 to 700 m, using two alternative meteorological data sets. The corresponding observed values ranged from 250 to 600 m. The authors concluded that the modelled results substantially depended on the analysis of the properties of the source term, especially regarding the convective heat fluxes from the fire.

The plume rise treatment of the BUOYANT model has also been evaluated against wind tunnel experiments (Martin et al., 1997; Kukkonen et al., 2000). These experiments addressed plumes rising into an inversion layer in the absence of wind. The study showed that the experimental and modelled final heights of plumes were in reasonable agreement. However, the predicted concentrations were slightly higher than those measured and the predicted plume radii were slightly shorter than the measured

ones; this indicates that there was slightly less entrainment occurring in the model than in the experiment. However, the perfectly calm conditions in the wind tunnel occur very seldomly in the atmosphere. Experimental data for non-isothermal conditions with wind would be required to evaluate the model in more frequently occurring conditions.

The dispersion module (after the plume rise regime) has also been tested against the Kincaid field trials (Olesen, 1995; Nikmo et al., 1999). Evaluation of the atmospheric dispersion module (Nikmo et al., 1999) showed a good agreement between the predicted and measured concentrations. However, for low concentrations, the model slightly underpredicted and for high concentrations slightly overpredicted the measured data.

Sofiev et al. (2012) has compared the BUOYANT plume rise predictions against the results of the Multi-angle Imaging SpectroRadiometer (MISR) Plume Height Project (e.g., Kahn et al., 2008). Sofiev et al. (2012) reported a tendency of the plume rise module to underestimate the experimental final plume rise. This could potentially be due to the missing latent heat contribution of the model.

In summary, in all of the above-mentioned evaluation studies of the plume rise and atmospheric dispersion modules of BUOYANT, the model predictions can be considered to have agreed well or fairly well with the observations.

The input data for the BUOYANT model has included various meteorological parameters and information on the properties of the fire and its surroundings (Kukkonen et al., 2014). However, a reliable determination of the required input data is a challenging task. This has up to date been the case also for all other models for the dispersion of buoyant plumes from major fires (e.g., Paugam et al., 2016, Lareau and Clements, 2017). The application of such models has therefore been possible only for expert users. In previous literature, no modules have been presented for the source term evolution of major fires. This has been a major obstacle to a more robust and widespread use of such plume rise models.

The overarching aim of this study has been to develop a comprehensive model for the dispersion of plumes originating from major fires, including all the relevant dispersion regimes. An objective has been to develop an operational and user-friendly model, which can be used also by well-trained meteorologists

and by trained emergency rescue personnel. We have therefore developed a novel semi-empirical module for predicting the initial conditions (i.e., source term) for models that treat the buoyant plume rise and the subsequent atmospheric dispersion of plumes from major fires. The input data required by the source term module is substantially simpler than the corresponding input required by the original model version.

The specific objectives of this study are the following. (i) The first objective is to present a new module for the initial properties of fire plumes in terms of fairly simple characterizations of major fires. We have also included the developed source term module in the BUOYANT model. (ii) The second objective is to compare the predictions of the latest version of the BUOYANT model against available field trials. (iii) The third objective is to present the structure and functioning of the developed operational emergency response model. Both the original research model and its operational application could in the future be used worldwide, the latter after some slight modifications, for better preparedness and rescue operations in case of major fires.

## 2    Model

For readability, we present first a brief overview of the whole modelling system. Second, we address the extension of the system to include the source term evolution.

### 2.1    Overview of the modelling system

The BUOYANT model is applicable for treating the initial plume rise and atmospheric dispersion of pollutants originating from buoyant sources. An overview of the modelling system is presented in the following. For a more detailed description of the mathematical model, the reader is referred to Kukkonen et al. (2014) and in the case of the atmospheric dispersion after the plume rise regime, to Nikmo et al. (1999).

The relevant flow regimes and an overview of the currently applicable modelling system have been presented in Figs. 1a-b. The model includes treatments (i) for the initial plume properties immediately above the fire (source term), (ii) for the dispersion of the buoyant plume and (iii) for the dispersion after the plume rise regime. All these modules constitute a model called BUOYANT. The model can be used to predict the spatial concentration distributions of pollutants originating from fires.

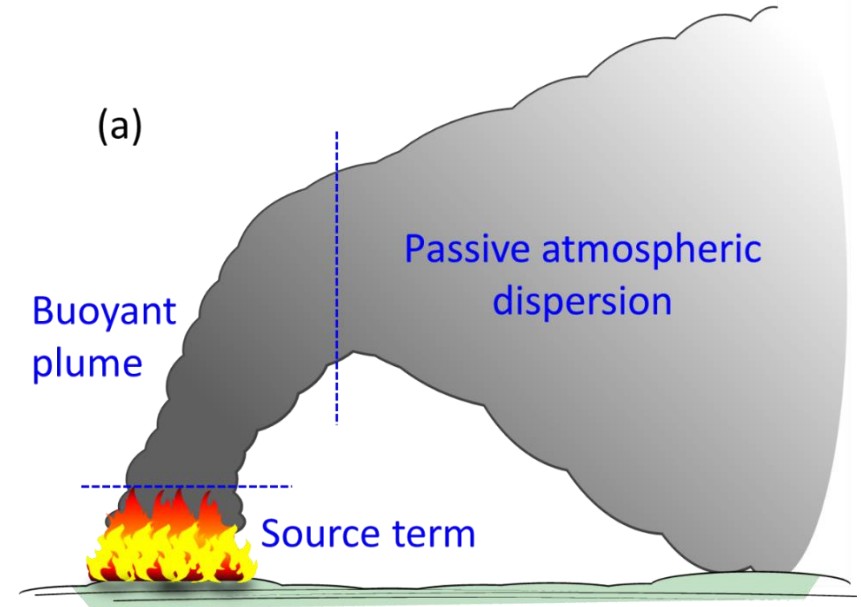


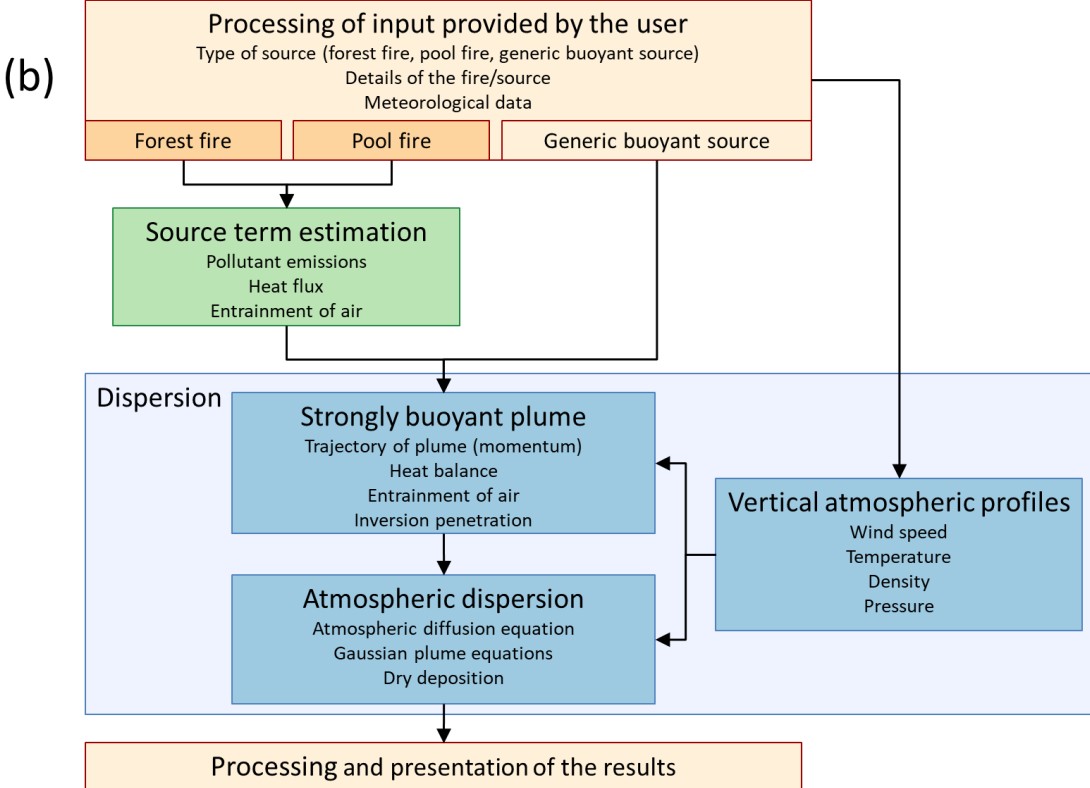

**Figure 1. (a) A schematic presentation of the flow regimes of the dispersion of a fire plume. (b) A block diagram of the BUOYANT model.**

The BUOYANT modelling system also includes an atmospheric dispersion module, which applies the gradient transfer theory and Gaussian equations in the vertical and horizontal directions, respectively (Nikmo et al., 1999). The source strength and the atmospheric conditions are assumed to remain constant in time during the atmospheric dispersion. However, after the plume rise regime, the plume properties can also be used as input information for any other dispersion model.


The information regarding the source term to be input to the plume rise module includes the following parameters: the radius of the (circular) source and its height above the ground, the temperature and the mass flux of the released mixture of contaminants and air, and the mass fraction of the released substance. In addition, one needs to know the molecular weight and heat capacity of the released substance

(Kukkonen et al., 2014).

The limitations of the BUOYANT model include: (i) The model assumes a steady state in terms of emissions and meteorology. However, the user can conduct multiple runs with various values of the emissions and meteorological parameters, to analyze the impacts of changing emissions and ambient

conditions. (ii) The current model version does not treat the impacts of phase changes of water in the plume (in particular, condensation and evaporation). (iii) The model adopts some values of model parameters according to the best available previous experimental and modelling studies. (iv) The chemical reactions of pollutants are not addressed.

The source term module has some additional limitations: (i) the source term module analyzes the amount of the burning material based on the properties of the living trees within the burnt area, but it does not explicitly include the pollution originating from burning litter, down-and-dead fine wood, shrub and herbaceous plants, (ii) the emissions of pollutants originating from the burning of wood are based on semi-empirical coefficients, determined in specific conditions, which may not be representative for all

conceivable forest fires, (iii) the intensity of burning in forest fires depends also on the climatic conditions, previous and current weather, the species of trees and other plants within the burning area, and the amount of water in the burning material; these factors are not explicitly addressed by the source term module.

The mathematical description of the plume rise and dispersion in the model contains three physical parameters, the values of which have been determined based on a comparison of model predictions and wind tunnel observations (Kukkonen et al., 2000). The parameters are the along and cross-plume air entrainment coefficients, and the so-called added mass term (which accounts for the plume having to push air out of the way as it rises). The values of these parameters have not been changed or adjusted in

any way after their initial determination. The model equations, therefore, do not contain free parameters that could be adjusted according to the measured values.

## 2.2    The extraction and pre-processing of meteorological data

The program can use either real-time or forecasted meteorological data produced by the numerical weather prediction (NWP) model HARMONIE; this model is run operationally at the FMI. In the

operational version of the BUOYANT model, this is the only option for providing the meteorological input data. However, in the research version of the BUOYANT model, the user can also either (i) manually provide the required meteorological input data, or (ii) use the meteorological data produced by another NWP model (such as, e.g., the HIRLAM model).

The acronym HARMONIE has been derived from "HIRLAM ALADIN Research on Mesoscale Operational Numerical weather processing In Euro-Mediterranean Partnership" (Nielsen et al., 2014). The modelling domain includes Fennoscandia, the Baltic Countries and the surrounding regions in the eastern Atlantic, Northern Central Europe and Russia.

The HARMONIE model was selected for three main reasons. First, this model has been thoroughly evaluated against experimental data and it is known to provide accurate, high-resolution weather forecasts for the whole of the modelling domain. The treatments of this model have been specifically adapted for the conditions in the Northern European region. Second, the NWP computations are performed operationally in-house, which simplifies the transfer of data between the operational program

and the NWP model. Thirdly, the HARMONIE system is the weather forecast model currently used in Finland; this model has replaced the previously used NWP model HIRLAM.

HARMONIE is a state-of-the-art NWP model, which has been widely used and developed in Europe. HARMONIE is a non-hydrostatic convection-permitting NWP model. The horizontal grid spacing of the

model is 0.022° (approximately 2.5 km). The vertical grid consists of 65 vertical hybrid levels. In this study, we applied the HARMONIE version cy40h11, which is in operational use at the FMI. Meteorological forecasts are continuously produced four times a day, with a temporal resolution of one hour and a forecast length of about two days ahead in time. A limitation of the HARMONIE model, as applied in the present study, is the limited geographic domain (only European and Atlantic regions).

Most of the meteorological variables required by the BUOYANT model are directly available from HARMONIE forecasts. The vertical structure of the atmosphere in the BUOYANT model is assumed to comprise three distinct layers: the atmospheric boundary layer (ABL), the capping inversion layer and the upper layer (Kukkonen et al., 2014). Variables that are readily available in HARMONIE include the height of the ABL and the vertical profiles of the temperature, pressure and wind speed.

However, the output data of this NWP model does not directly include some of the input values required by the BUOYANT model. We have therefore constructed a continuously functioning pre-processor model, which predicts the required meteorological parameters based on the output of the HARMONIE model. These parameters are the ambient temperature and pressure, the lateral wind components, the inverse Monin-Obukhov length, the height of the atmospheric boundary layer and the vertical profiles of temperature and wind speed above ABL.

In the ABL the vertical variations of wind speed and temperature are in this study assumed to be described with profiles based on the Monin-Obukhov similarity theory, as presented by Kukkonen et al. (2014). Monin-Obukhov length is estimated based on the values of the turbulent momentum stress near the ground surface, as forecasted by HARMONIE. The two-layered thermal structure above ABL (inversion and upper layers) is analyzed by applying the HARMONIE predictions with a method modified from Fochesatto (2015).

In the upper layer (above the inversion layer), the wind speed is assumed to be constant (representing the geostrophic flow), whereas within the inversion layer the wind speed is assumed to change with a constant gradient from its value at the top of the ABL to the geostrophic value. The constant geostrophic wind speed was assumed to be equal to the arithmetic mean of HARMONIE forecasts between the top of the inversion layer and the height of 5 km.

The meteorological roughness length, which will be needed in the atmospheric dispersion computations, is extracted from the CORINE (CoORrdination of INformation on the Environment) land cover information in 2012, using, in addition, the weighting coefficients modified by Venäläinen et al. (2017).

## 2.3 The source term module

The source term module can be used to analyze the characteristics of the plume generated by a fire, which will be needed for the subsequent computations on the evolution of a buoyant plume. The source term module includes treatments for the properties of a fire plume just above the flame tips, based on information on the characteristics of the fire. The module has been designed to be used also for operational purposes; we have therefore tried to keep the amount and nature of the input data as limited and simple as possible.

The current module version can be applied for two significant categories of fires, viz. the forest and liquid pool fires. In the case of forest fires, the input data of the source term module has been selected to include (i) the area of the forest on fire, (ii) the number of trunks per unit area of forest, (iii) the average height of trees and (iv) the average bole diameter at breast height. The input data in the case of liquid pool fires include (i) the burning substance, and (ii) the surface area of the liquid pool on fire. The results of the source term module include (i) the mass fluxes of gaseous and particulate matter produced by a fire, the mass flux of entrained air, and (ii) the characteristic scales of radius, temperature and vertical velocity of the plume.

The source term module does not include a treatment of the propagation of the actual fire in the terrain; models have been developed for this purpose in other studies (Sullivan, 2009a-c). The influence of the wind also has not been included in the analysis of the source term, although it is included for the overall BUOYANT model. In the case of intensive fires under prevailing light or moderate wind speeds, this is a reasonable assumption. In the case of very high wind speeds, allowing for the influence of the wind would increase the dilution of the source term, compared with the present computations. The influence of the vertical wind structure in the atmosphere has been allowed for in the treatment of the buoyant plume and naturally, in the subsequent atmospheric dispersion (Kukkonen et al., 2014). The fire is

assumed to be in the flaming stage; that is the fire burning regime that produces the highest atmospheric concentrations of pollutants.

In the following, we will first address the modelling of (i) the heat fluxes generated by fires and (ii) the average heights of the flames. These results will subsequently be used for deriving equations for (iii) the
radius, velocity, temperature and (iv) molar flux of a fire plume.

### 2.3.1 Heat fluxes generated by fires

The heat energy of fire is mostly convected or radiated from the fire region (e.g., Heskestad, 1984 and 2016). A smaller fraction of the heat will also be conducted into the ground (e.g., Ichoku and Kaufman, 2005; Freeborn et al., 2008; Ichoku et al., 2012). The so-called theoretical heat release rate (i.e., heat
energy flux) from the fire can be defined to occur if the burning material is completely combusted. The theoretical heat release rate ($Q$) can be evaluated as (Heskestad, 2016)

$$Q = q_{m,f} H_c,$$

(1)

where $q_{m,f}$ is the mass burning rate (i.e., rate of mass burned per time); and $H_c$ is the lower heat of
complete combustion (heat energy per burned mass). The lower heat of combustion refers to a situation, in which the fire products are in the state, in which they have been formed (e.g., Drysdale, 2016), i.e., the potential subsequent phase transitions and chemical reactions have not been considered. For instance, in this situation, any liquid water in the fuel that has been vaporized during the burning process is assumed to be in vapour form. The possible condensation of water, therefore, does not by definition
contribute to the heat released by the fire.

The combustion efficiency $\chi$ is defined as the ratio of the total to the theoretical heat release rate. This efficiency is close to unity for some fire sources (e.g., methanol and heptane pools), but it may deviate significantly from unity for others (Heskestad, 2016). However, as this efficiency is not commonly
known in operational applications, we have assumed for simplicity complete combustion ($\chi = 1$).

The heat flux generated by a fire ($Q$) is propagated in the form of convection ($Q_c$), radiation ($Q_r$) and by other ($Q_o$) means (e.g., conduction),

$$Q = Q_c + Q_r + Q_o \equiv \varepsilon_c Q + \varepsilon_r Q + \varepsilon_o Q, \tag{2}$$


where $\varepsilon_c$, $\varepsilon_r$ and $\varepsilon_o$ are the fractions of convective, radiative, and other heat fluxes of the total heat flux $Q$, respectively (by definition, $\varepsilon_c + \varepsilon_r + \varepsilon_o = 1$). Laboratory experiments on biomass burning have demonstrated values of $\varepsilon_o \sim 0.35$ (Freeborn et al., 2008). For large fires, the radiative fraction $\varepsilon_r$ tends to decrease with the increasing size of the fire (Heskestad, 2016).


The convective heat flux $Q_c$ can be written simply as (e.g., Achtemeier et al., 2012; Kukkonen et al., 2014; Heskestad, 2016)

$$Q_c = c_p \rho u \pi r^2 \Delta T, \tag{3}$$

where $c_p$ is the specific heat capacity of the plume, $\rho$ is the density of the plume, $u$ is the characteristic velocity of the plume, $r$ is the characteristic radius of the plume and $\Delta T = T - T_a$ is the excess temperature of the plume. $T$ is the characteristic temperature of the plume and $T_a$ is the temperature of ambient air.

### 2.3.2 Mean flame height

The flame intermittency, $I(z)$, is defined as the fraction of time, during which part of the flame is above
the height $z$ (Heskestad, 2016). The flame intermittency decreases with height; it is equal to unity at the fire source and vanishes at sufficiently large heights. The mean flame height ($\lambda$) is defined as the altitude, at which the flame intermittency has declined to half of its initial value. At the height $\lambda$, most of the combustion reactions have taken place, and at higher altitudes, the plume can therefore be considered to be inert with a fairly good accuracy (Heskestad, 2016).


Several experimentally derived correlations have been proposed for $\lambda$. As would physically be expected, $\lambda$ has been found to correlate positively with the fire Froude number, i.e., dimensionless heat release rate $Q^*$ and the pool diameter $d$; which can be written as $\lambda \sim d\, Q^*$ (e.g., Luketa and Blanchat, 2015). In general, the Froude number is a dimensionless number defined as the ratio of the flow inertia to the external field; in many applications the external field is gravity. Grove and Quintiere (2002), Dupuy et al. (2003),

Luketa and Blanchat (2015) and Heskestad (2016) have presented comparisons of several of the correlations for $\lambda$ in terms of experimental data and against each other.

In this study, we have adopted the correlations of $\lambda$ presented by Zukoski et al. (1985). These correlations performed amongst the best ones in a comparison of predictions and experimental data in large-scale liquid natural gas (LNG) burner experiments that were reported by Luketa and Blanchat (2015). The correlations of Zukoski et al. (1985) also provided physically sensible results in conceivable fire scenarios. These correlations can be written as

$$\lambda = 3.30dQ^{*2/3}, \text{if} \qquad Q^* < 1,$$
$$\lambda = 3.30dQ^{*2/5}, \text{if} \qquad Q^* \geq 1,$$
(4a-b)

where $d$ is the diameter of a fire source or an equivalent diameter for a noncircular fire with the same area, and $Q^*$ is the Froude number of the fire, defined here as

$$Q^* = \frac{Q}{\rho_a c_{pa} T_a (gd)^{1/2} d^2},$$
(5)

where $\rho_a$ is the density of ambient air, $c_{pa}$ is the specific heat capacity of air at constant pressure and $g$ is the acceleration due to gravity. According to Eq. (5), the Froude number is large ($Q^* \gg 1$) for fires, in which the energy output is relatively large compared to its physical diameter.

### 2.3.3 Radius, velocity and temperature of a fire plume

The source term module presented in this study is based on the buoyant plume equations commonly called the Morton-Taylor-Turner model (Morton et al., 1956; hereafter referred to as MTT). The MTT model is applicable for a steady plume of buoyant material that rises vertically in a calm (no wind), unstratified atmosphere. The MTT model applies to point sources that have a relatively small difference in density compared to ambient air density (Boussinesq approximation), or to a region above the source, in which air entrainment has brought the plume density sufficiently close to the ambient value.

The coupled first-order differential equations of the MTT model govern the evolution of the characteristic plume radius ($r$), velocity ($u$) and density deficit ($\Delta\rho = \rho_a - \rho$) above a point source. For readability, we have presented these equations and their mathematical solution in Appendix A.

As shown in Appendix A, the solution of these equations can be written in terms of the convective heat flux $Q_c$ (e.g., Heskestad, 1998 and 2016),

$$r = \frac{6\alpha}{5}z, \tag{6a}$$

$$u = \frac{5}{6}\left(\frac{9}{10\pi\alpha^2}\right)^{1/3} g^{1/3}\left(c_p\rho_a T_a\right)^{-1/3} Q_c^{1/3} z^{-1/3}, \tag{6b}$$

$$\frac{\Delta\rho}{\rho_a} = \frac{5}{6}\left(\frac{9\pi^2\alpha^4}{10}\right)^{-1/3} g^{-1/3}\left(c_p\rho_a T_a\right)^{-2/3} Q_c^{2/3} z^{-5/3}, \tag{6c}$$

where $\alpha$ is a dimensionless entrainment coefficient.


The entrainment assumption of the MTT model states that the velocity of entrained air across the plume edge ($u_e$) is proportional to the plume velocity

$$u_e = \alpha u. \tag{7a}$$

However, there are many instances, in which plumes can not be modelled according to the Boussinesq approximation. Observations indicate that in the non-Boussinesq case, the entrainment velocity depends on the ratio of the plume density and the ambient density (Ricou and Spalding, 1961). Morton (1965) suggested, based on experimental evidence and theoretical considerations, an additional proportionality of the entrainment velocity,


$$u_e = \alpha \left(\frac{\rho}{\rho_a}\right)^{1/2} u, \tag{7b}$$

which is usually referred to as the Ricou-Spalding entrainment model; hereafter referred to as RS. Equation (7b) indicates reduced entrainment into lighter plumes, in comparison with entrainment to plumes of near ambient air density.


For extending the model to other than point sources, a concept of virtual source can be introduced. The virtual source is located below the actual area source and is accounted for by replacing the height $z$ by $z - z_{vs}$, where $z_{vs}$ is the height of the virtual source. In addition, to accommodate large density deficiencies, Morton's extension of the MTT model results in that the plume radius in the right-hand side of Eq. (6a) has to be multiplied by a factor $(\rho_a/\rho)^{1/2}$ and the ratio $\Delta\rho/\rho_a$ in Eq. (6c) has to be replaced by $\Delta\rho/\rho$ (Morton, 1965).


The comparison of model predictions and measurements of fire plumes above the flames have to a large extent supported the use of the above-described theory (Heskestad, 1984 and 1998). However, the prediction accuracy can be improved by using experimentally adjustable coefficients. The plume radius ($r_{\Delta T}$), and the mean values of velocity ($u_0$) and excess temperature ($\Delta T_0 = T_0 - T_a$) at the plume centre line have been found to obey the following relations (Heskestad, 1984 and 1998):


$$r_{\Delta T} = C_1 \left(\frac{T_0}{T_a}\right)^{1/2} (z - z_{vs}), \tag{8a}$$

$$u_0 = C_2 \left(\frac{g}{c_p \rho_a T_a}\right)^{1/3} Q_c^{1/3}(z - z_{vs})^{-1/3}, \tag{8b}$$

$$\Delta T_0 = C_3 \left(\frac{T_a}{g c_p^2 \rho_a^2}\right)^{1/3} Q_c^{2/3}(z - z_{vs})^{-5/3}, \tag{8c}$$

where $r_{\Delta T}$ is the plume radius at the point where the excess temperature is $0.5\Delta T_0$, $T_0$ is the mean temperature in the centerline of the plume, and $C_1 = 0.12$, $C_2 = 3.4$ and $C_3 = 9.1$ are experimental coefficients. The above values of the coefficients $C_i$ ($i = 1,3$) are based on experimental investigations of heated air jets and large-scale pool fires (George Jr. et al., 1977; Kung and Stavrianidis, 1982). For burn experiments in rack storage fires, Kung et al. (1988) and Tamanini (2010) have recommended slightly
different values of $C_i$ ($i = 1,3$).

For fire sources, which do not have substantial in-depth combustion, the height of the virtual source $z_{vs}$ can be estimated based on the experimental relation of Heskestad (1984),

$$z_{vs} = -1.02d + 0.083\left(\frac{Q}{1000}\right)^{2/5}.$$

(9)


A fire source does not have substantial in-depth combustion if a major fraction of the volatiles released (2/3 or higher) undergo combustion above the fuel array (Heskestad, 1984). Fire sources with substantial in-depth combustion include, e.g., very openly constructed, or well-ventilated wood cribs.

### 2.3.4   Molar flux of a fire plume

Assuming ideal gas behaviour, the molar flux of gaseous material ($q_n$) through a circular source plane is

$$q_n = \frac{p_a}{R_g T}\pi r^2 u,$$

(10)

where $p_a$ is the atmospheric pressure (pressure within the plume is assumed to be equal to the ambient value) and $R_g$ is the molar gas constant. The flux $q_n$ comprises the molar fluxes of air ($q_{n,a}$) and gaseous

combustion products ($q_{n,c}$),

$$q_n = q_{n,a} + q_{n,c}.$$

(11)

Detailed modelling of the two selected application areas has been presented in Appendix B.

### 2.3.5   Interface of the source term and plume rise regimes

The mean velocity ($u_0$) and excess temperature ($\Delta T_0$) in Eqs. (8b-c) are values at the centre line of a fire plume. As the radial distance from the centre line increases, $u$ and $\Delta T$ approach their ambient values. We have assumed Gaussian distributions for the radial distributions of velocity and temperature in the source term regime.

However, the subsequent modelling phase in the plume rise regime assumes that the plume is described by a uniform (top-hat) distribution of physical quantities. If the centerline values of the Gaussian

distributions ($u_0$, $\Delta T_0$) would be used to represent a top-hat profile, the convective heat would not in general be conserved at the boundary of the source term module and the plume rise module.

We have therefore presented a solution in Appendix C, in which the convective heat is conserved at the boundary of the two modelling regimes. According to this solution, the top hat velocity ($u$) and excess temperature ($\Delta T$) in the plume rise regime are related to the values at the centre line of the Gaussian plume in the source terms regime as follows (utilizing the values of the coefficients $C_i$ ($i = 1,3$) presented in Eqs. (8a-c)):


$$u = 0.86u_0, \tag{12a}$$
$$\Delta T = 0.83\Delta T_0. \tag{12b}$$

The model is not applicable for $\pi C_1^2 C_2 C_3 < 1$, as required by the conservation of heat energy, as presented in detail in Appendix C.

Detailed modelling of the fluxes of compounds originating from the two selected fire types has been presented in Appendix B.

### 2.3.6   Summary of the use of the source term module equations and the computer implementation

For both types of fires, mean flame height is computed from Eqs. (4a-b). The initial plume radius, vertical centerline velocity, and the excess temperature of the plume are evaluated from Eqs. (8a-c) and (9). The

mass flux of entrained air is computed applying Eqs. (10-11).

In the case of a forest fire, the convective heat flux and the mass fluxes of pollutants originating from a fire are computed based on Eqs. (1), (B6), and (B7), and applying the emission factors presented by Kaiser et al. (2012). In the case of a pool fire, the convective heat flux and the mass fluxes of pollutants

originating from a fire are computed based on Eqs. (1), (3), (B1), and (B2). In addition, in the case of pool fires, the module applies the fuel property data presented in Table B1. The velocity and temperature required for the subsequent plume rise regime are computed based on Eqs. (12a-b).

## 3 The operational version of the model

The main aim of developing the operational version of the model was to provide a user-friendly tool of assessment for various emergency response personnel. The operational model can be used for emergency contingency planning and the training of emergency personnel, in case of forest and pool fires. The stakeholders and users of the operational model currently include a wide range of emergency response personnel in Finland, the operational meteorologists at the Finnish Meteorological Institute (FMI), experts at the Ministry of the Interior in Finland and researchers at the FMI.

### 3.1 Overview and functioning of the operational model

The operational program has been named FLARE (Fire pLume model for Atmospheric concentrations, plume Rise and Emissions). An overview of the model structure has been presented in Fig. 2. This model contains the program BUOYANT for conducting the physical and chemical computations, a graphical user interface, and various modules for processing the input and output data of the model. The operational version can be used remotely via an internet connection.

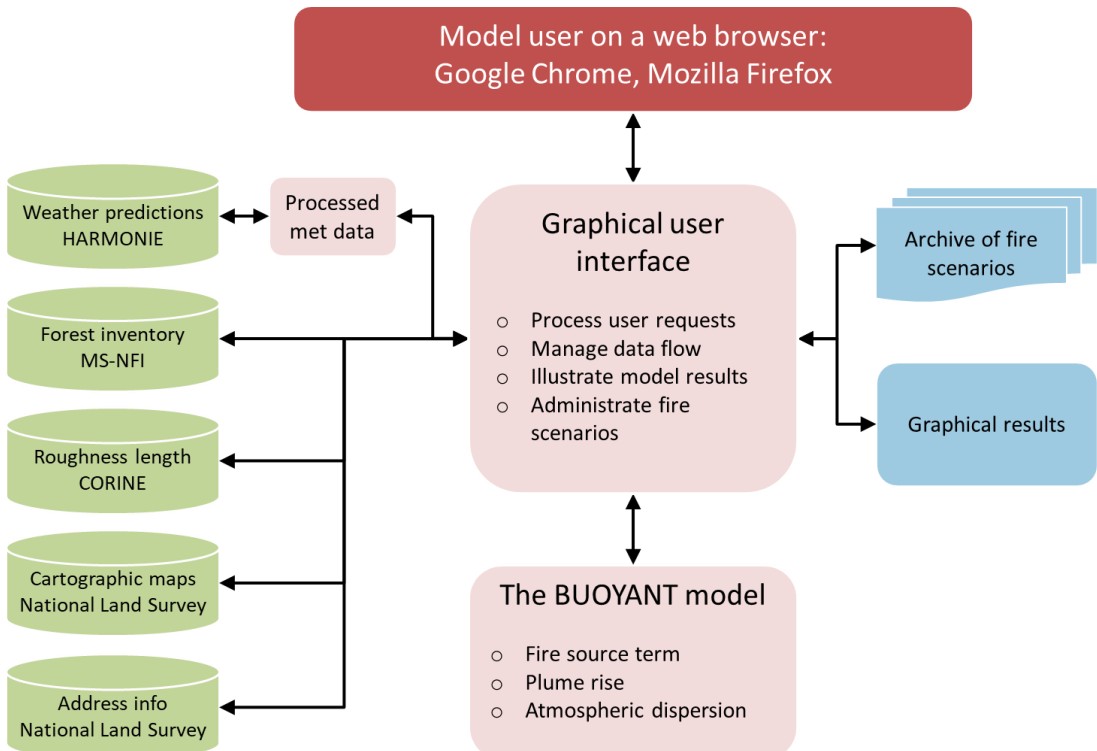

**Figure 2. Block diagram of the operational model. The red, green, light red and blue colours indicate the internet connections, the data sources, the main modules and the model output,**

**respectively. The arrows indicate the flow of data. Acronyms: MS-NFI = Finnish Multi-Source National Forest Inventory, CORINE = CoORrdination of INformation on the Environment.**

The graphical user interface provides the necessary input to the operational model. If desired, the user can change the weather and forest information already provided for the model. Visualization of results
and management of archived fire events can also be done via the user interface. It is recommended that the operational model will be used with the latest versions of the most commonly used browsers, such as Google Chrome or Firefox.

The operational model addresses forest fires and liquid pool fires. The user should only indicate the place
and time of the fire, the estimated size of the fire area and the substance released in the event of a pool fire. In addition, the model pre-processes and provides for the computations of three main types of input data: meteorological parameters, forest information and geographic maps.

The program will subsequently check that all the user-specified input data values and their combinations
are physically reasonable. The program will also check that the computations address cases, which are within the applicability range of the operational model. In case of unrealistic or unreasonable input values, the program will either request the user to confirm the value or to input a more realistic value.

However, the current version of the operational model can be used only for locations that are situated in
Finland, or close vicinity of the country. The operational model could be extended to function also in other countries and regions, by expanding especially the cartographic and forest inventory datasets. In case of missing input datasets, the model could also be modified to skip some of the input processor modules and ask the user to input the corresponding values. For instance, if there would not be a suitable forest inventory available for the considered domain, the user would be asked to supply the required
information on the characteristics of the forest.

The user can archive descriptions of fire events, which contain input data for a range of potential fire scenarios. These cases can then be retrieved, edited as necessary, and used for further computation.

The program presents the numerical results as pollutant iso-concentration curves on maps. The current operational version presents spatial concentration distributions of carbon dioxide ($CO_2$) near the ground level.

Both the operational version of the model (FLARE) and the original research model (BUOYANT) use
the same code for describing the dispersion and transport of a buoyant plume. The program of the FLARE model includes the whole of the BUOYANT program, and all the core physics computations in the operational model are done using the research code. However, there is a dedicated user interface in the operational model, to facilitate easier use. The BUOYANT model allows versatile post-processing of the model results, whereas the FLARE model includes post-processing in a standard format. The FLARE
model can be accessed by the commonly used web browsers, whereas the BUOYANT model has been designed to be as computer platform-independent as possible.

An example application of the operational model has been presented in Appendix D.

### 3.2     The pre-processing of the input datasets in the operational model

The functioning of the operational model has been made as user-friendly as possible, by an automatic pre-processing of several input materials. The program can use either real-time or forecasted meteorological data produced by the numerical weather prediction (NWP) model HARMONIE (as described in more detail in section 2.2). The meteorological parameters and forest inventory data will be extracted and preprocessed for the spatial coordinates and the time specified by the model user. The
operational model also presents the results on geographic maps, for the domain selected by the user.

The automatic online use of weather and forest data makes the use of the operational model substantially quicker and simpler. This will also reduce potential human errors. For non-expert users, the determination of the required meteorological variables would otherwise be a very challenging task. In case of long-term
fires, the user can also use the forecasted meteorological values, for forecasting the spread of the fire plumes up to two days ahead in time.

These input datasets and pre-processors are briefly described in the following.

### 3.2.1 Forest information

In the case of forest fires, the amount of burning material is modelled based on a national inventory of forests (Mäkisara et al., 2019). This inventory has been compiled by the Natural Resources Institute Finland, and it is called the Multi-Source National Forest Inventory of Finland (MS-NFI). The methods and results of this inventory have been presented by Tomppo and Halme (2004). The inventory is publicly available.

The inventory contains forest resource maps, classified under 44 themes. The inventory has been constructed based on satellite images, field data collected nationally, and digital map data. The data covers the whole of Finland and part of the areas in Northern Sweden and Russia. The spatial resolution of the data is 16 m × 16 m.

The operational model uses the results of this inventory, corresponding to the year 2015. The relevant parameters are the number of trunks per unit area of forest, the average height of trees, and the average bole diameter at breast height. The values of these parameters are automatically selected from the inventory, corresponding to the user-specified location and surface area of the fire.

### 3.2.2 Geographic map information

The operational model provides as output spatial concentration distributions near the ground level, presented on digital maps. The model uses open-access maps provided by the National Land Survey of Finland (2021). The user can specify the location of the accident by simply clicking that point on the map, specifying the geographic coordinates or by writing the street address. The accident location will then be seen on the map as the logo of the FMI.

## 4    Evaluation of the BUOYANT model against experimental data

We have evaluated the model against the experimental data of the Prescribed Fire Combustion and Atmospheric Dynamics Research Experiment, RxCADRE. The properties of the fire source term could be analyzed using the source term module that has been presented in this article. However, for this particular measurement campaign, it is more accurate to use directly the values that were reported in the

database and the relevant publications regarding the experiment. We have therefore evaluated the BUOYANT model mostly by excluding the source term module presented in this study.

For comparison purposes, we have also done computations using the source term module developed in
this study. However, this comparison includes a major challenge. Most of the trunks of the trees were not burnt in the RxCADRE experiments, although part of the down-and-dead fine wood was burnt. On the other hand, the operational BUOYANT model assumes that the trunks of the trees were burnt. This is a reasonable assumption in an operational model, as it has been primarily developed for describing intensive major fires. The comparison of the numerical results of the BUOYANT model including the
source term module with measured RxCADRE data is nevertheless useful to illustrate the potential use of the operational model in case of minor forest fires. We have also done one model run regarding the sensitivity of the modelled results on the fire intensity.

## 4.1   Overview of the RxCADRE experiments

The experiments were designed to collect extensive data before, during, and after the active burning
periods of prescribed fires. These datasets are one of the most comprehensive field campaigns to date, providing accurate measurements of various aspects of the fires, including meteorology, the evolution of fires, their energy, emissions, and airborne concentrations (Ottmar et al., 2016a; Clements et al., 2019; Prichard et al., 2019).

The RxCADRE experiments were conducted in Florida, USA during 2008–2012. These datasets have previously been used, for instance, in the evaluation of the empirical-stochastic plume model Daysmoke (Achtemeier et al., 2012), the Stochastic Time-Inverted Lagrangian Transport (STILT) model (Mallia et al., 2018), and the Weather Research and Forecasting Model combined with a semi-empirical fire-spread algorithm (WRF-SFIRE) (Mallia et al., 2020; Moisseeva and Stull, 2019).


For readability, we present a brief description of the experiments and measured data, which were used in this study. For a more detailed description of the experiments, the reader is referred to a special issue of the International Journal of Wildland Fire, which was aimed to document the RxCADRE study (Peterson and Hardy, 2016). In particular, Ottmar et al. (2016a and 2016b) has presented overviews of the
RxCADRE study.

The RxCADRE measurement campaign consisted of six smaller (less than 10 ha) and ten larger (10 - 900 ha) prescribed fires (Ottmar et al., 2016a). Measured data that is sufficient for dispersion model evaluation is available for three larger fires: two grass fires (named by the authors as L1G and L2G, in which L presumably refers to larger experiments and G to grass) and one sub-canopy forest fire (named as L2F, in which F refers to forest) (Clements et al., 2016; Dickinson et al., 2016; Strand et al., 2016). In the case of these three experiments, data is available regarding the meteorological conditions, fire emissions and airborne concentrations.

As the main focus of the present study was on forest fires, we have selected the L2F fire for model evaluation. This experiment was conducted over a burning block of size 151 ha at Eglin Air Force Base, Florida. The L2F burn experiment was conducted on 11 November 2012, from 18 to 22 UTC. The amount of burned organic material was approximately 4.13 Mg ha$^{-1}$ (Ottmar et al., 2016b). Approximately 65 % of the burned material consisted of litter, 21 % of down-and-dead fine wood ($\leq$ 7.6 cm in diameter), 13 % of shrub, and about 1 % of herbaceous plants (grasses and forbs). The experimental data is publicly available in the Research Data Archive of the United States Department of Agriculture (RDA, 2018). The dataset includes aircraft concentration measurements of CO, $CO_2$, $CH_4$, and $H_2O$ (Urbanski, 2014b; Strand et al., 2016). However, there were neither aircraft measurements on the particulate matter nor ground-based measurements in the publicly available dataset. We have therefore conducted the model evaluation using the concentration data on gaseous substances from aircraft measurements in the experiment L2F.

The accuracy of the meteorological measurements within the L2F burn was analyzed by Seto and Clements (2015a and 2015b). They concluded that the data are qualitatively reliable. Urbanski (2014b) analyzed the accuracy of the airborne concentrations, regarding the positioning by GPS and the applied spectroscopic methods (Cavity ring-down spectroscopy, CRDS). The estimated analytical uncertainty of the CRDS measurements varied from 1 % to 1.5 % for $CO_2$ and $CH_4$, and from 2 % to 15 % for CO.

## 4.2   The selected experiment, the sub-canopy forest fire (L2F)

The L2F burn consisted of a sub-forest-canopy fire over a 1.51 km$^2$ plot (approximately a rectangle of size 1.6 km $\times$ 1.0 km). The burn was ignited with drip torches mounted on all-terrain vehicles (ATVs)

moving along three parallel firing lines of approximately 1.5 km in length, which transected the plot from the northeast to the southwest (Hudak et al., 2017; Ottmar et al. 2016). The firing lines were ignited simultaneously between 18:02:53 UTC and 18:27:36 UTC, i.e., the ignition process took almost 25 min (Hudak et al., 2017).


The radiative heat fluxes of the fire were measured using the long-wave infrared (LWIR) technique (Dickinson et al., 2016). The measurements were done on board a twin-engine Piper Navajo aircraft, which was used to make repeated passes at about three-minute intervals (Dickinson et al., 2016). Pixel values of LWIR imagery were calibrated to total ground-leaving radiative exitance (defined as the radiant

flux emitted by a surface per unit area), also termed fire radiated flux density (FRFD) (Dickinson et al., 2016). Fire radiative power (FRP) of a calibrated LWIR imagery pixel is the product of the pixel area (1.5 m × 1.5 m) and its measured FRFD. FRFD data is available from the Research Data Archive (Hudak et al., 2016a). The total FRP of the fire at any given time is the sum of all the pixels on fire. Fire pixels were separated from nonfire pixels using as a cut-off value the estimated background radiative flux

density of 1070 W m$^{-2}$ (Hudak et al., 2016b).

The aircraft measurements onboard a Cessna 337 aircraft yielded data on fresh emissions, the vertical profile of the smoke, plume height and the dispersion of smoke (Urbanski, 2014b). Measurements were conducted at distances of up to 25 kilometers downwind from the source. Measurements were done as a

so-called parking garage and corkscrew flight profiles. Parking garage vertical profiles involved short (approximately 10 km) horizontal transects, roughly perpendicular to the axis of the smoke plume, taken at multiple elevations. Corkscrew profiles were centered on the plume downwind from the burn unit. Parking garage and corkscrew maneuvers were designed for measuring the horizontal and vertical concentration distributions, respectively.


The measured temporal evolutions are presented in Fig. 3 for the FRP, the burning area, and their averages in the course of the fire.

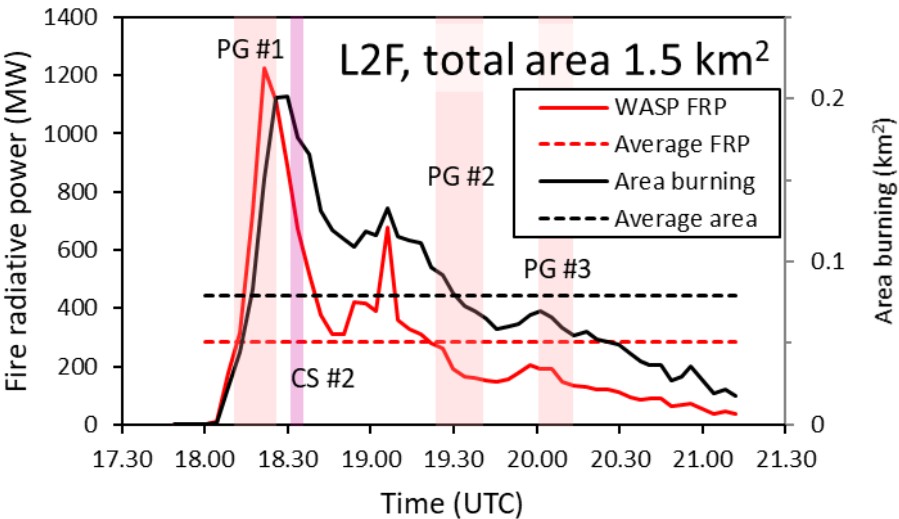

**Figure 3. Measured temporal evolutions of the fire radiative power (FRP) (red curve, left-hand vertical axis) and the active fire area (black curve, right-hand vertical axis) of the L2F burn. The FRP values were taken from the data presented by Hudak et al. (2016a). WASP refers to Wildfire Airborne Sensor Program. Transects of the measurement aircraft are denoted by four vertical bars coloured light red and violet (marked as PG #1, CS #2, PG #2 and PG #3); data extracted from Urbanski (2014b). PG and CS refer to the parking garage and corkscrew flight maneuvers, respectively. Time-averaged values of FRP and the active fire area are shown with dashed red and black lines, respectively.**

We have compiled the information in Fig. 3 based on the data by Hudak et al. (2016a) and Urbanski (2014b). The figure illustrates that there were substantial temporal changes in the fire during the experiment, both regarding the intensity of the fire and its spatial extent. The timing of the aircraft flights focuses on one hand on the most intensive periods of the fire (PG #1 and CS #2), on the other hand on covering the whole period of the fire (PG #2 and PG #3). Flight CS #1 was done before the ignition of the fire.

The measured airborne concentrations represented average values during two seconds; these are included in the available dataset. However, we have computed and applied observed and modelled concentrations as 20 seconds centered moving averages, in agreement with Mallia et al. (2018).

## 4.3 Analysis of the meteorological variables and the fire source

### 4.3.1 Analysis of the meteorological variables

The vertical profiles of wind speed, temperature and pressure required by the model were determined by applying the onsite measurements (Clements et al., 2016). The meteorological measurement campaign of RxCADRE consisted of a variety of measurement platforms and instrument types. We have applied the data collected by the California State University - Mobile Atmospheric Profiling System (CSU-MAPS) for the determination of atmospheric properties. For evaluations near the ground surface (up to a height of 30 m), we used measurements in the micrometeorological mobile tower, which was located downwind of the L2F burn plot. Above the height of 30 m, Doppler lidar measurements were used. The lidar was placed on the perimeter of the L2F burn unit.

We applied the meteorological measurements before the ignition of the L2F burn. Ten-minute averages of measured wind speed, wind direction, ambient temperature and ground-level air pressure were applied for further processing. The Monin-Obukhov length ($L$) was estimated by fitting the atmospheric vertical profiles used in the BUOYANT model (Kukkonen et al., 2014) to the averaged temperature and wind measurements, using the method presented by Nieuwstadt (1978). The two-layered thermal structure above the atmospheric boundary layer (ABL) was analyzed by applying the measured temperature profile, according to the method by Fochesatto (2015). The experimental and modelled atmospheric vertical profiles of temperature, wind speed and wind direction are shown in Fig. 4.

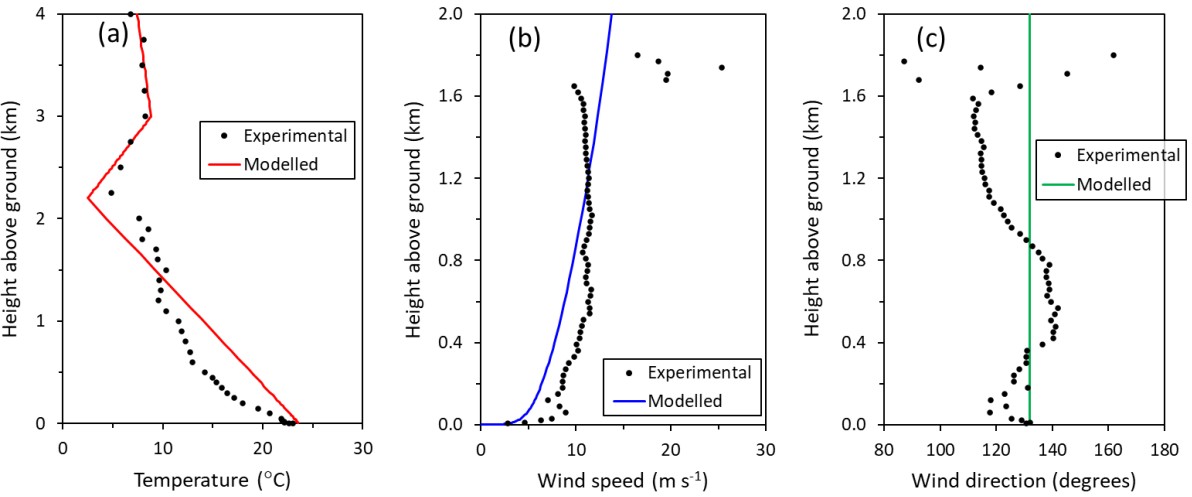

**Figures 4a-c. Experimental (black circles) and modelled (solid red, blue and green lines) vertical profiles of (a) temperature, (b) wind speed, and (c) wind direction.**

The prevailing wind direction was measured as south-easterly at 132°. The modelled wind speed and ambient temperature at the altitude of 10 m were analyzed to be 3.2 m s$^{-1}$ and 24 °C, respectively. The atmospheric stability was estimated to be moderately stable ($L^{-1}$ = 0.0011 m$^{-1}$). Based on the observed temperature profile, the height of the ABL was estimated to be 2.2 km. The gradients of potential temperature were estimated to be 0.0193 K m$^{-1}$ and 0.0094 K m$^{-1}$ within the inversion and upper layers, respectively. Wind speed above the ABL was assumed to be constant and equal to the modelled value at the top of the ABL, 14 m s$^{-1}$. A summary of the required meteorological input quantities of the model is presented in Table 1.

Table 1. Summary of the meteorological information required by the BUOYANT model related to the L2F burn of the RxCADRE experimental campaign. References to applied experimental data (where applicable) are presented in the column 'Derived from'.

| Quantity | Derived from | Value |
|---|---|---|
| **The inverse of Monin-Obukhov length ($L^{-1}$)** | Estimated by fitting atmospheric vertical profiles used in the model to the averaged temperature and wind measurements (Seto and Clements, 2015a and 2015b). The fitting is based on the method presented by Nieuwstadt (1978). | 0.00108 m$^{-1}$ |
| **Meteorological roughness length ($z_0$)** | Estimated by fitting the vertical wind profile used in the model to the averaged wind measurements (Seto and Clements, 2015a and 2015b). | 0.190 m |
| **Heat roughness length** | Assumed to be equal to $z_0$. | 0.190 m |
| **The temperature at ground level** | The measured ambient temperature at the ground level (Seto and Clements, 2015a). | 296.169 K |
| **Ambient pressure at ground level** | Measured ambient pressure at the ground level (Seto and Clements, 2015a). | 101930.1 Pa |
| **Reference height ($z_r$).** | Height at which wind speed is specified. | 10 m |
| **Wind speed at $z_r$** | Assessed with profiles based on the Monin–Obukhov similarity theory, applying the estimated $L$, friction velocity and $z_0$. | 3.22 m s$^{-1}$ |

| | | |
|---|---|---|
| **Wind direction** | Temporally (10 min) and spatially (up to a height of 1 km) averaged wind direction observations (Seto and Clements, 2015a and 2015b). | 132° |
| **Mixing height ($h$)** | Based on the measured temperature profile (Seto and Clements, 2015a and 2015b). | 2200 m |
| **Top of the inversion layer** | Analyzed by applying the measured temperature profile (Seto and Clements, 2015a and 2015b), according to the method by Fochesatto (2015). | 3000 m |
| **A potential temperature gradient within the inversion layer** | Same as "Top of inversion layer". | 0.0193 K m$^{-1}$ |
| **The potential temperature gradient in the upper layer** | Same as "Top of inversion layer". | 0.0094 K m$^{-1}$ |
| **Wind speed in the upper layer** | Assumed to be equal to the modelled value at $h$ (height of the ABL). | 14.26 m s$^{-1}$ |


### 4.3.2 Analysis of the properties of the fire source

We have assumed a steady-state of the fire in the modelling. The following properties of the fire source need to be known as input values for executing the BUOYANT model: (i) the fire radiative power (FRP), (ii) the convective fraction of energy released from the fire ($\varepsilon_c$), (iii) the physical extent of the fire ($a_f$),

(iv) the temperature of fire products ($T_f$), and (v) the mass fluxes of the fire products ($q_f$).

For evaluating the properties from (i) to (iv), we have applied the observed values as reported by Hudak et al. (2016a) and Jimenez and Butler (2016), averaged over the measurement period. Time-averaged mass flux (v) of $CO_2$ was derived from the experimentally determined fuel consumption and emission

factors (Hudak et al., 2016b; Strand et al., 2016). The averaged values of the fire source applied for the dispersion computations were: FRP = 283 MW, $\varepsilon_c$ = 0.324, $a_f$ = 0.079 km$^2$, $T_f$ = 58 °C, and $q_f$ = 113 kg s$^{-1}$ (the mass flux of $CO_2$). The time-averaged FRP and $a_f$ are also shown in Fig. 3.

A summary of the required fire source input quantities of the model is presented in Table 2.


Table 2. Summary of the fire source information required by the BUOYANT model related to the L2F burn of the RxCADRE experimental campaign. References to applied experimental data are indicated in the column 'Derived from'.

| Quantity | Derived from | Value |
|---|---|---|
| Source radius ($r$) | Average based on the LWIR images (Hudak et al., 2016a). | 158.58 m |
| Emission height ($z_e$) | Assumed. | 1 m |
| Source temperature ($T$) | Average of the measured flame temperatures (Jimenez and Butler, 2016) | 330.85 K |
| Total mass flux ($q$; air and contaminant) | Eq. (3), where $q = \rho u \pi r^2$; $T$ is the average of the measured flame temperatures (Seto and Clements, 2015a); $T_a$ is the measured ambient temperature at $z_e$ (Seto and Clements, 2016a); and the average $Q_c$ is related to the measured average radiative heat flux ($Q_r$), i.e. average FRP (Fig. (3)), as $$\begin{cases} Q_c = \varepsilon_c Q \\ Q = Q_c + Q_r \end{cases} \Rightarrow Q_c = \frac{\varepsilon_c}{1-\epsilon_c} Q_r,$$ where $Q$ is the average heat generated by the fire (Eq. (2)); and $\varepsilon_c$ is the average of the experimentally determined fraction of convective heat flux of $Q$ (Jimenez and Butler, 2016). It is assumed here that $Q$ is propagated only through convective and radiative processes (i.e. $Q_o = 0$ in Eq. (2)). | 3876.51 kg s⁻¹ |
| Mass fraction of contaminant ($\chi$) | $\chi = q_c q^{-1}$, where $q_c$ is the average of the mass flux of $CO_2$ estimated with $$q_c = a F_c EF_{CO2} t_b^{-1},$$ where $a$ is the total area burned determined from the LWIR images (Hudak et al., 2016a).; $F_c$ is the amount of fuel consumed by the fire per unit area; $EF_{CO2}$ is the emission factor of $CO_2$ (in units mass of $CO_2$ per mass of dry fuel consumed); and $t_b$ is the duration of the burn. For $F_c$ and $EF_{CO2}$ we have applied the averages of the experimentally derived estimates presented by Hudak et al. (2016b) and Strand et al. (2016), respectively. The duration of the burn $t_b$ was determined from the LWIR images (Hudak et al., 2016a). | 0.0290562 |

We have also conducted one additional model run, regarding the impact of the changes in input data on the properties of the fire. The fire properties for the base case represent averages over three hours (Table 2 and Fig. 3). For the additional model run, we have chosen the averages of the fire properties representing the early high-intensity (HI) stage during the first hour of the burn (from 18:04:26 to

19:05:56). A summary of the required fire source input quantities of the model for the HI stage is
presented in Table 3. For easier comparison, the input data of the source module for the base case (Table
2) is reproduced in Table 3. The average FRP for the HI phase (535 MW) was 90 % higher compared to
the base case (283 MW).

Table 3. Summary of the fire properties applied for the modelling of the base case and the case that uses
input data measured during the high-intensity stage of the fire (the average values over both periods).
Notation: HI = high-intensity fire during the first hour of the L2F burn.

| Quantity | Average over the entire burn | Average over the HI stage |
| --- | --- | --- |
| Source radius (m) | 158.58 | 193.1 |
| Emission height (m) | 1 | 1 |
| Source temperature (K) | 330.85 | 335.65 |
| Total mass flux (kg s$^{-1}$) | 3876.51 | 6442.54 |
| Mass fraction of contaminant (-) | 0.0290562 | .00326859 |

Although the source term module could not be convincingly evaluated using the RxCADRE experiment,
we have also compared the predictions of the BUOYANT model obtained using the source term module.
The main reason for the source term module not being suitable for the RxCADRE experiment is that the
standing tree trunks were not burnt during the experiment. However, the comparison could be useful to
illustrate the potential use of the operational FLARE model (i.e., the operational model) in cases, in
which the forest fire does not burn the standing tree structures.

The required input data for the source term module (cf. section 2.2, paragraph three) was not documented.
Therefore, the required input regarding the properties of the standing trees was assessed visually from
the pictures taken from the L2F burn (Jimenez and Butler, 2016). The (i) average number of trunks per
unit area of forest, (ii) average height of trees, and (iii) average bole diameter at breast height were
estimated to be 625 ha$^{-1}$, 12.7 m and 29 cm, respectively. The size of the fire was assumed to be equal to
the size (radius) assumed for the base case (Table 2).

### 4.4   Results of the model evaluation

We consider in the following the observed and modelled excess concentrations, i.e., concentrations
subtracted by background concentration. These represent the contributions of the fire. We focus on the

comparison of the measured and predicted spatial concentration distributions, in the horizontal and vertical directions. The aircraft measurements were specifically designed to measure such distributions. The modelling, both including or excluding the source term module, does not contain any free parameters and has not been adjusted to the measured data in any way.

In the BUOYANT model, the model makes the transition from the plume rise regime to the dispersion modelling regime when there is no buoyancy left in the plume (Kukkonen et al., 2014). This distance and the corresponding height substantially depend on the area and intensity of the fire, and the meteorological conditions. In the RxCADRE model simulation presented here, the plume rise stage terminated at a distance of 6.2 km from the source, at a height of 510 m (the height at the plume centerline).

Measured and modelled vertical excess concentrations of $CO_2$ are presented in Fig. 5, for three parking garage and the second corkscrew flight paths. The flight duration varied from 7 (CS #2) to 16 minutes (PG #1 and #2). In the case of the corkscrew flight manoeuvre (Fig. 5b), the results have been presented for three modelling options: (i) the base case, using estimated input data for an average period of three hours, excluding the source term module ("Modelled, Ave, no ST"), (i) the modelling based on the measured input data during the first high-intensity hour of the fire, excluding the source term module ("Modelled, HI, no ST") and (iii) the modelling for the base case, but including the source term module ("Modelled, ST").

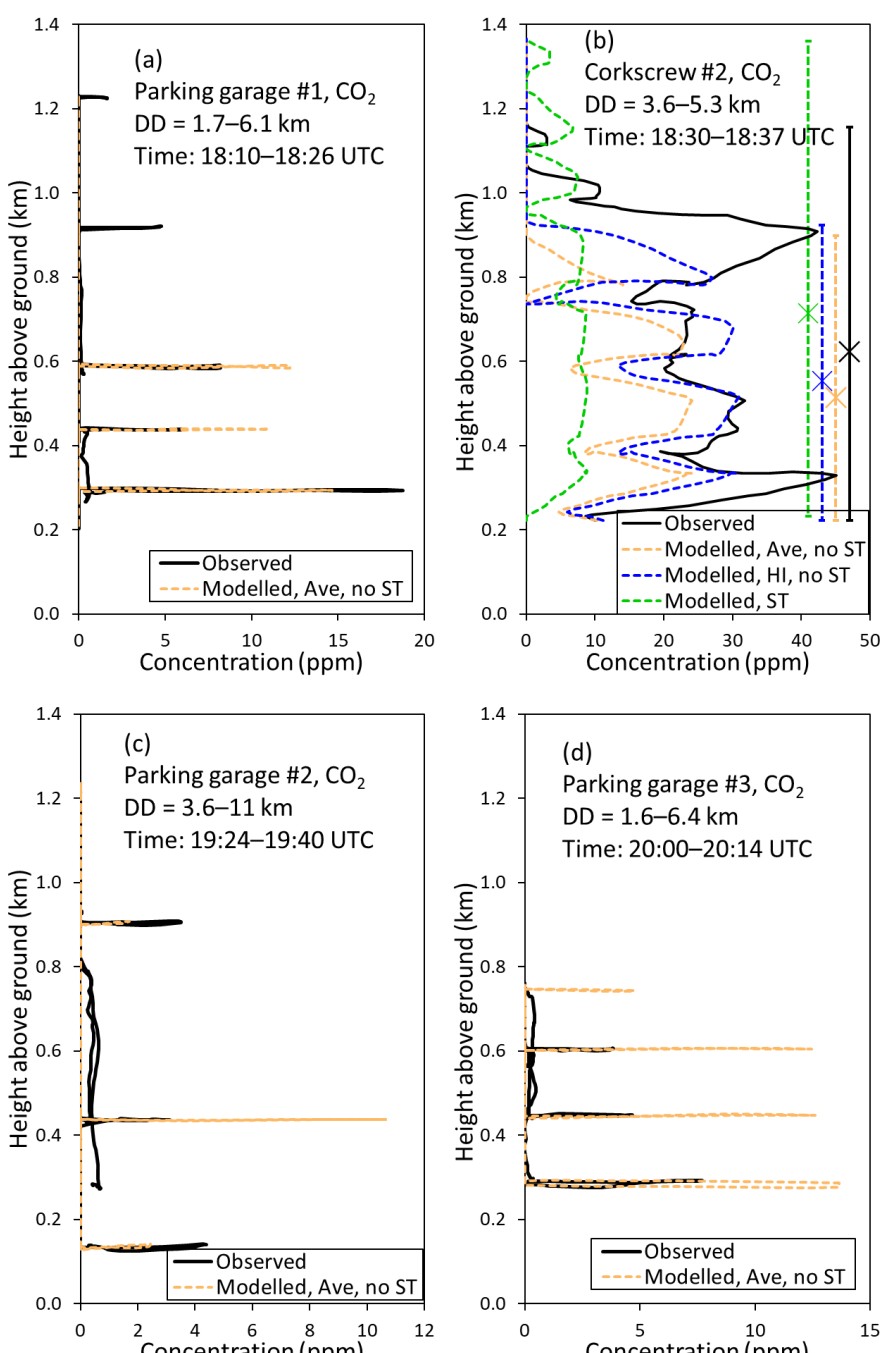

**Figures 5a-d. The vertical excess concentrations of CO₂ versus height for three parking garage and the second corkscrew flight manoeuvres. The curves represent observed data (solid black), modelled base case (dashed orange), modelled values using the input data during the high-intensity (HI) phase of the fire (dashed blue), and modelled values for the base case, including the source term (ST) module (dashed green). In Fig. 5b, the observed and modelled vertical extents of plume have also been shown as vertical bars on the right-hand side of that panel. The centres of mass of the plumes are marked with cross-hairs. Notation: DD = the downwind distance from the centre of the L2F burn block, time = the time interval of the manoeuvre of the measurement aircraft, UTC = Coordinated Universal Time.**

805

810

815

In the case of the parking garage flight manoeuvres (Figs. 5a, c and d), the observed vertical excess concentration distributions of $CO_2$ against the height were on average reasonably well captured by the model. However, there were both over- and underpredictions of the highest measured concentrations. For instance, regarding the parking garage flight #1, the highest measured peak concentration was underpredicted, whereas the second and third highest measured concentrations were overpredicted. For parking garage flight #3, the magnitudes of all the highest peaks were overpredicted.

In the case of the corkscrew number 2 manoeuvre (Fig. 5b), the modelling using the input data corresponding to the initial high-intensity period of the fire ("Modelled, HI, no ST") agreed best with the measured data. This result was to be expected, as corkscrew flight number 2 was conducted during the first hour of the fire. This agreement was better than for the other two modelling options regarding both (i) the magnitude and variation of the concentrations with height, and (ii) for the height of the centre of mass of the fire plume. The concentrations for the base case ("Modelled, Ave, no ST") were, as expected, clearly lower than those for the "Modelled, HI, no ST" case.

For the modelling option "Modelled, HI, no ST", the model estimated the plume to be at a slightly lower elevation compared to the observations: the modelled centre of mass of the distribution is approximately at a height of 554 m, while measurements indicate an elevation of 624 m. The modelled vertical extent of the plume (701 m) was smaller than indicated by the measurements (932 m).

The base case computations excluding the source term module ("Modelled, Ave, no ST") were based on directly measured experimental data regarding the source term properties (i.e., properties over the flame tips of the fire). Taking into account the substantial uncertainties in deriving the source term data and the relevant meteorological data, together with the uncertainties of the plume rise modelling, we consider these measured and predicted results to be reasonably well in agreement.

For the corresponding computations using the source term module ("Modelled, ST"), the properties above the flame tips were computed based solely on the properties of the burning forest material. The input data of the source term module included the estimated average diameters, heights and areal density of the tree trunks. The source term module then proceeds to estimate, at which rate the tree material is

burned (kg s$^{-1}$ m$^{-2}$), and the pollutant fluxes produced by the burning. Due to these model assumptions, the source term module is best applicable for analyzing highly intensive major forest fires, in which a substantial fraction of the tree trunks will be burnt.


For this specific experiment, the underlying assumptions of the source term module were not completely applicable, as the burning material included litter, down-and-dead fine wood, shrub and herbaceous plants. The results have nevertheless been presented for this case, to illustrate the operational model performance in the case of smaller and less intensive fires. It is expected that using the directly measured

source term properties (the cases marked as "no ST") will be more accurate than modelling these based solely on forestry data.

For the base case, including the source term module, the concentrations were lower than for the comparable base case, excluding the source term module, by a factor of two slightly more. The

computation including the source term module also predicted a wider vertical distribution of the fire plume. Compared with the measured concentration values, the values using directly measured source term properties ("Modelled") were expectedly relatively in better agreement. The vertical extent of the plume was somewhat underpredicted and overpredicted by excluding and including the source term module, respectively; neither of these modelling options was better in that respect.


Measured and modelled (for the base case) horizontal distributions of the excess concentrations of $CO_2$ are presented in Figs. 6a-c during the three parking garage flight paths. The DD and time values in the panels of the figures are the distances and measurement times of the aeroplane; the distances were measured from the fire location during the whole measurement period of that flight. Each of the

horizontal measurements, as shown by each curve in the panels, represents shorter periods than indicated in the flight measurement time values.

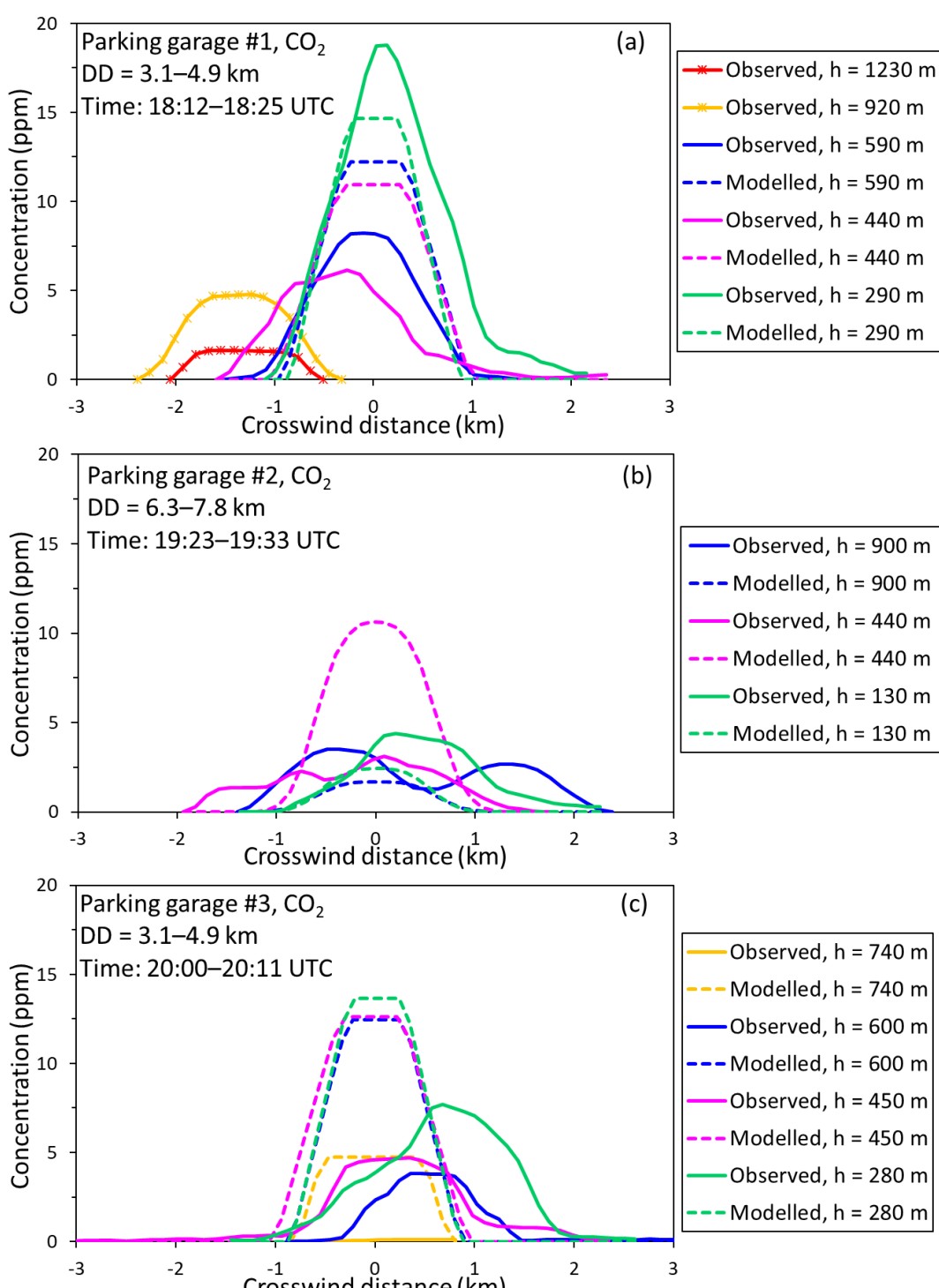

**Figures 6a-c. Observed (solid lines) and modelled (dashed lines) horizontal distributions of the excess concentrations of CO₂ for three parking garage flight manoeuvres. The corresponding measured and modelled curves have been presented with the same colour. Notation: DD = downwind distance from the centre of the L2F burn block, time = the time interval of the manoeuvre of the measurement aircraft, *h* = the elevation above ground, UTC = Coordinated Universal Time.**



Overall, the model predictions compare fairly well with the observations for all the three considered flights, as presented in Figs. 6a-c. However, the concentration peaks tend to be overestimated by the model for most cases, and the measured widths of the plume are underestimated by the modelling.

There are at least two physical reasons for both the over-prediction of peaks and the under-prediction of the plume widths. First, the fire was ignited along three multiple approximately straight lines, whereas the model assumes a uniform and temporally constant fire strength over the source area. In the modelling, the area of the burning has also been assumed to be circular, whereas in the experiments, the burning area is variable in time, and its shape is not exactly circular. These differences between the modelling

and the experiments tend to cause a more diffuse plume in the experiments, compared with the modelling setup.

Second, all of the flights PG #1 and PG #3, and half of the flight PG #2 occurred within the plume rise regime in the model. The concentration distribution in this modelling regime is represented by a top-hat

profile. In reality, the horizontal profile was probably more diffuse, resulting in a wider measured concentration distribution and lower peak concentrations.

The modelling has assumed a steady-state of the fire and the meteorological conditions. In particular, the fire intensity was assumed to be temporally constant throughout the whole duration of the experiment.

The model, therefore, tends to underpredict the concentrations in the initial stages of the fire (PG #1 and CS #2) and overpredict these in the later stages (PG #2 and #3). In the later stages of the fire, the observed concentration distributions corresponded to larger values of the crosswind distances, compared with the corresponding predicted distributions, at all the considered elevations, as presented in Fig. 6c. This effect could have been caused by a temporal turn of the wind direction in the later stages of the burn.


For the modelling of fire plumes, it is crucial to predict sufficiently accurately the vertical structure of the atmosphere, especially the potentially existing temperature inversions. Kukkonen et al. (2014) previously compared the predictions of the BUOYANT model against the measurements in two other field measurement campaigns. They found that, e.g., analyzing the meteorological conditions in the

SCAR-C experiments (Kaufman et al., 1996; Hobbs et al., 1996; Gassó and Hegg, 1998) using two

different meteorological methods resulted in substantially different meteorological input data values for the model.

In the case of the RxCADRE measurements, the relevant meteorological parameters have been carefully measured and well reported. However, the application of such datasets in determining vertical atmospheric profiles of the relevant quantities, and the atmospheric stability conditions, will result in some degree of inaccuracy in the dispersion modelling.

## 5   Conclusions

We have presented a refined version of a mathematical model, BUOYANT, which has been designed for analyzing the formation and dispersion of plumes originating from major fires. The model addresses the average properties along a trajectory (also called the cross-plume integrated properties) of a rising plume in a vertically varying atmosphere. The model also considers the impacts on plume rise of possibly occurring inversion layers (Kukkonen et al., 2014).

A major limitation of currently available plume rise models has been that such models require input data on the fire, which are very challenging to determine reliably, such as the mass and heat fluxes from the fire. In other words, there have been no modules for predicting the early evolution of the fire (i.e., the fire source term). In the present study, we have suggested a novel module for this purpose, and the BUOYANT model has been extended to include this module. The use of such a source term module could potentially result in more accurate predictions of the plumes of major fires, and increase the understanding of the initial fire processes, and their influence on the input values for the modules describing the plume rise regime. The new model extension also facilitated a design of an operational model version, which can be used in a much more straightforward way, compared with the use of the original research model.

The source term module uses as input the information on the characteristics of the fire, and it is used to model the properties of a fire plume just above the flame tips. The current version of the source term module can be applied to two significant categories of fires, viz. the forest and liquid pool fires. In future work, the source term module could be generalized to address also other fire types. In the case of forest fires, the input data of the source term module includes the area of the forest on fire, the number of trunks

per unit area of forest, the average height of trees and the average bole diameter at breast height. The main structure of the source term module is based on the differential equations for releases of buoyant material, which govern the evolution of the plume radius, velocity and density difference. The module is semi-empirical, as it also relies on various experimental results on fire plumes.


The main limitations of the BUOYANT model include the following. The model assumes a steady state in terms of emissions and meteorology. The current model version does not treat the impacts of phase changes of water in the plume. The chemical reactions of pollutants are not addressed during the source term and plume rise stages. The source term module has additional limitations. The source term module

predicts the amount of the burning material based on the properties of the living trees within the burnt area, but it does not explicitly include the pollution originating from other burning materials. The prediction of the emissions of pollutants is based on semi-empirical coefficients, determined in specific conditions, which may not be representative of all conceivable forest fires. In reality, the intensity of burning in forest fires depends also on the climatic conditions, previous and current weather, the species

of trees and other plants within the burning area, and the amount of water in the burning material.

We have compared the predictions of the refined BUOYANT model against the experimental field-scale data of the RxCADRE campaign (Prescribed Fire Combustion and Atmospheric Dynamics Research Experiment). Overall, the predicted concentrations of $CO_2$ fairly well agreed with the aircraft

measurements of the RxCADRE campaign. However, regarding the vertical distributions of the concentrations during the parking garage flight manoeuvres, there were both over- and underpredictions of the highest measured concentrations.

Regarding the vertical distributions of the concentrations during the corkscrew number 2 manoeuvre, we

have predicted the results using three modelling options. These were (i) using experimentally measured input data for the fire properties during the whole duration of the fire, (ii) using experimentally measured input data during the high-intensity period of the fire, and (iii) using the source term module for predicting the fire properties. This particular flight manoeuvre was done during the high-intensity stage of the fire. The predictions agreed best with the measurements, as expected when we used the directly measured fire

properties for the high-intensity period. The concentrations using the source term module, which applied

solely the forestry data regarding the fire, but not the directly measured fire properties, were somewhat under-predicted.

We also evaluated the model performance for predicting the horizontal distributions of the $CO_2$ concentrations. Overall, the model predictions fairly well compared with the observations for all the three considered flights. However, the concentration peaks tended to be moderately overestimated by the model for most cases, and the measured widths of the plume were underestimated by the modelling. The physical reasons for these deviations were related (i) to the geometry and ignition of the experimental fires, which did not completely correspond to the assumptions in the modelling, and (ii) to the simplified assumptions on the form of the concentration distributions in the model and (iii) possibly, a temporal turn of the wind direction in the later stages of the burn.

The fire intensity was assumed to be temporally constant throughout the whole duration of the experiment. Due to the steady-state assumption, the modelling tended to underpredict the concentrations in the initial stages of the fire and to overpredict these in the later stages. The source term module currently assumes that the burned material consists solely of standing tree trunks. Clearly, on one hand, also other kinds of plant material contribute to the burning in a forest, and on the other hand, especially in less intensive fires, tree trunks may be only partially burnt. In the future development of fire source term modules, these factors should be described in more detail.

Another source of uncertainties in the modelling is the prediction of the relevant meteorological parameters. The meteorological measurements in the RxCADRE campaign have been carefully measured and reported. However, the application of such data for determining vertical atmospheric profiles of the relevant quantities, and the atmospheric stability conditions, will inherently result in some inaccuracies.

The operational version of the model is a user-friendly tool of assessment that can be used by various emergency response and rescue personnel. This model can be used for emergency contingency planning and the training of emergency personnel, in case of forest and pool fires. The model has been used by the Finnish rescue authorities up to date. However, it would be possible to use both the original research model and its operational application also worldwide, the latter after adding the necessary cartographic

material. This would be expected to result in improved preparedness and better, knowledge-based rescue actions in case of major fires.


**Appendix A. The Morton-Taylor-Turner model for a buoyant plume**

Let us consider a plume from a point source, assuming no momentum flux at the source, uniform ambient air density and the Boussinesq approximation. The conservation of mass, momentum and buoyancy can be written as (Morton et al., 1956)


$$\frac{d}{dz}(r^2 u) = 2r u_e, \qquad \text{(volume/mass),} \qquad \text{(A1a)}$$

$$\frac{d}{dz}(r^2 u^2) = r^2 g\left(\frac{\rho_a - \rho}{\rho_a}\right), \qquad \text{(momentum) and} \qquad \text{(A1b)}$$

$$\frac{d}{dz}\left(r^2 u g \frac{\rho_a - \rho}{\rho_a}\right) = 0, \qquad \text{(buoyancy),} \qquad \text{(A1c)}$$

where $z$ is the height above ground; $r$ is the radius of the plume; $u$ is the vertical velocity of the plume; $u_e$ is the rate of entrained air across the plume edge (entrainment velocity); $g$ is the acceleration due to gravitation; $\rho_a$ is the density of ambient air; and $\rho$ is the density of the plume.


The entrainment velocity is assumed to be proportional to some characteristic velocity at height $z$ (Morton et al., 1956)

$$u_e = \alpha u, \qquad \text{(A2)}$$

where $\alpha$ is an experimentally defined proportionality constant (the entrainment constant) relating the entrainment velocity to the vertical velocity within the plume. Equation (A2) is often referred to as the Morton-Taylor-Turner entrainment model.

The solution of Eqs. (A1a-A1c) is (Morton et al., 1956)


$$r = \frac{6\alpha}{5} z, \qquad \text{(A3a)}$$

$$u = \frac{5}{6\alpha}\left(\frac{9}{10}\alpha B\right)^{1/3} z^{-1/3}, \qquad \text{(A3b)}$$

$$g\frac{\rho_a - \rho}{\rho_a} = \frac{5B}{6\alpha}\left(\frac{9}{10}\alpha B\right)^{-1/3} z^{-5/3},$$ (A3c)

where the constant buoyancy flux $B$ is

$$B = r^2 u g \frac{\rho_a - \rho}{\rho_a}.$$ (A4)

Assuming ideal gas behaviour, the buoyancy flux can be written in terms of convective heat flux ($Q_c$) (Heskestad, 2016)

$$B = \frac{gQ_c}{\pi c_p T_a \rho_a},$$ (A5)

where $c_p$ is the specific heat capacity of air at constant pressure; and $T_a$ is the temperature of ambient
air.

## Appendix B. Detailed modelling of the selected application areas

Semi-empirical modules of mass burning rate are presented in the following for liquid pool and forest fires. Mass fluxes of the emitted chemical compounds (e.g., CO, $CO_2$) from the fire are, by definition, determined employing the modelled mass burning rate and emission factors.

### B1. Mass fluxes of pollutants originated from liquid pool fires

Hottel (1959) suggested how to analyze liquid pool burning, according to heat transfer principles (Babrauskas, 1983). According to Hottel (1959), the mass burning rate is governed by the heat exchange between the flames and the pool surface. The heat exchange mechanisms are (a) radiant flux from the flames into the pool, (b) convective flux from the flames into the pool, (c) re-radiant heat loss ($Q_{rr}$) due to the high temperature of the pool and (d) conduction losses and non-steady terms ($Q_{misc}$).

The term $Q_{rr}$ is commonly small, and quantitative expressions for $Q_{misc}$ are usually not available (Babrauskas, 1983). For simplicity, the terms $Q_{rr}$ and $Q_{misc}$ are therefore customarily ignored (Babrauskas, 1983). Hottel (1959) analyzed the experimental data of Blinov and Khudiakov (1957), concluding that two burning regimes are possible: radiatively dominated burning for larger pools and convectively dominated burning for smaller pools. The distinction between the two regimes can be drawn at the pool diameter of approximately 0.2 m (Hottel 1959; Babrauskas, 1983; Chatris et al., 2001). For fire hazard analysis, liquid pool fires will rarely be significantly dangerous, if they are smaller than about 0.2 m in diameter (Babrauskas, 2016). It is therefore commonly necessary to treat only pool burning in the radiative regime.

Zabetakis and Burgess (1961) suggested (cf. Babrauskas, 1983; Chatris et al., 2001; Brambilla and Manca, 2009), based on the work of Hottel (1959), the following relationship to represent the mass burning rate ($q_{m,f}$) of a liquid pool in the radiative dominated regime:

$$q_{m,f} = q_{m,\infty} A \left(1 - e^{-k\beta d}\right), \tag{B1}$$

where $q_{m,\infty}$ is the mass burning rate per unit area of an infinite-diameter pool; $A$ is the surface area of the burning liquid pool; $k$ is the extinction coefficient of the flame; $\beta$ is a mean-beam-length corrector; and $d$ is the diameter of pool. For small $d$ the flames are said to be optically thin, while for larger $d$ the

flames become optically thick (Babrauskas, 1983). For optically thick flames, a further increase in $d$ does not result in a corresponding increase in back radiation into the pool. Such attenuation is accounted for by the coefficient $k\beta$ (Brambilla and Manca, 2009).

Values of the empirical coefficients $q_{m,\infty}$ and $k\beta$ for a variety of fuels have been proposed by, for instance, Babrauskas (1983), Rew et al. (1997) and Chatris et al. (2001). The surface area of the burning liquid pool and the name of the liquid fuel have to be provided as input data for the module.

Experimental values of the lower heat of complete combustion ($H_c$) for various burning fuels (liquids)
have been tabulated by, for example, Babrauskas (1983), McGrattan et al. (2000) and Hurley (2016).

The total heat generated by a liquid pool fire is here assumed to be propagated only through convective and radiative processes, i.e., $\varepsilon_o = 0 \Leftrightarrow \varepsilon_c = 1 - \varepsilon_r$. Radiometer measurements from large fire experiments involving different combustible liquids (such as crude oil, heptane and kerosene) suggest that the
radiative fraction ($\varepsilon_r$) decreases with increasing fire diameter ($d$), according to (McGrattan et al., 2000)

$$\varepsilon_r = \varepsilon_{max} e^{-kd}, \tag{B2}$$

where $\varepsilon_{max} = 0.35$; and $k = 0.05$ m$^{-1}$.

Yield ($y_i$) of a fire product $i$ is defined as the ratio of the mass generation rate of $i$ to the mass burning rate of the combusting fuel (e.g., Tewarson, 1980; Khan et al., 2016)

$$y_i = \frac{m_{w,i} q_{n,i}}{q_{m,f}}, \tag{B3}$$

where $m_{w,i}$ is the molecular weight of species $i$; and $q_{n,i}$ is the molar flux of species $i$. Experimental values
of yields under well-ventilated fire conditions have been listed by, for instance, Ross et al. (1996) and Hurley (2016).

Molar fluxes ($q_{n,i}$) of gaseous fire products are calculated using Eq. (B3). Finally, the molar flux of air can be calculated utilizing Eqs. (10) and (11).

Examples of fuel property data are shown in Table B1.

Table B1. Examples of the fuel property data (Babrauskas, 1983, Hurley, 2016). The quantities $H_c$, $q_{m,\infty}$ and $k\beta$ are defined in the text, and $y_{CO2}$, $y_{CO}$, $y_{hc}$ and $y_s$ are the yields of $CO_2$, $CO$, hydrocarbons and soot, respectively. Dashes indicate that either there are no measurements, or their values are less than 0.001.

| Material | $H_c$ | $q_{m,\infty}$ | $k\beta$ | $y_{CO2}$ | $y_{CO}$ | $y_{hc}$ | $y_s$ |
|---|---|---|---|---|---|---|---|
| | kJ kg$^{-1}$ | kg (m$^2$ s)$^{-1}$ | m$^{-1}$ | g g$^{-1}$ | g g$^{-1}$ | g g$^{-1}$ | g g$^{-1}$ |
| Acetone ($C_3H_6O$) | 25800 | 0.041 | 1.9 | 2.14 | 0.003 | 0.001 | 0.014 |
| Benzene ($C_6H_6$) | 40100 | 0.085 | 2.7 | 2.33 | 0.067 | 0.018 | 0.181 |
| Butane ($C_4H_{10}$) | 45700 | 0.078 | 2.7 | 2.85 | 0.007 | 0.003 | 0.029 |
| Heptane ($C_7H_{16}$) | 44600 | 0.101 | 1.1 | 2.85 | 0.01 | 0.004 | 0.037 |
| Kerosene | 43200 | 0.039 | 3.5 | 2.83 | 0.012 | 0.004 | 0.042 |
| LNG (mostly $CH_4$) | 50000 | 0.078 | 1.1 | 2.72 | – | – | – |
| LPG (mostly $C_3H_8$) | 46000 | 0.099 | 1.4 | 2.85 | 0.005 | 0.001 | 0.024 |

## B2. Mass fluxes of pollutants originated from forest fires

McAllister and Finney (2016a and 2016b) have measured the mass burning rate of wildland fires. Wood cribs, such as the one presented in Fig. (B1), have been used in fire testing. Block (1971) developed a theoretical model of the crib burning rate. Heskestad (1973) combined the experimental results of Gross (1962) and Block (1971) with the theoretical findings of Block (1971). This resulted in a relation of the mass burning rate ($q_{m,f}$) on the porosity ($\phi$) of the crib (see also McAllister and Finney, 2016a),

$$\frac{10^3 q_{m,f}}{A_s b^{-1/2}} = f(\phi),$$ (B4)

where $A_s$ is the exposed surface area of the sticks in the crib; and $b$ is the thickness of the sticks (defined in Fig. B1). The functional form of $f$ was found to be approximately (McAllister and Finney, 2016a)

$$f(\phi) = 1 - e^{-50\phi}. \tag{B5}$$

For well-ventilated cribs or loosely-packed porous burning cribs, $\phi$ is large and $f(\phi)$ approaches unity.

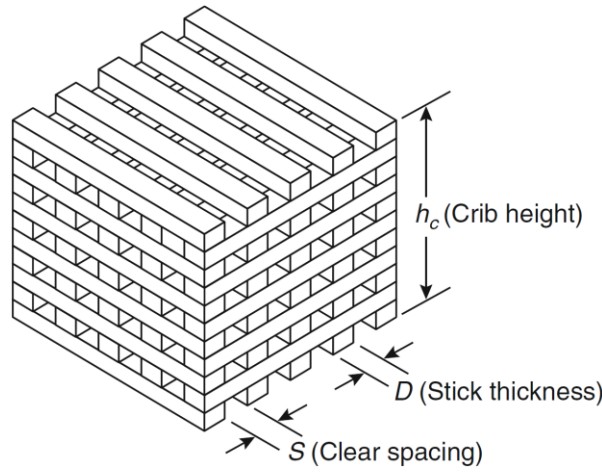

**Figure B1. The general arrangement of a wood crib (Babrauskas, 2016).**


Wildland fuel beds are commonly porous, i.e., the porosity is large (McAllister and Finney, 2016a). However, according to Tang (2017), both well-ventilated and under-ventilated fires can exist in forested regions. We have assumed in this study, for simplicity, that the fuel beds are porous ($f(\phi) = 1$).

Let us define the diameter of a tree trunk at human breast height, $d_{bh}$. Commonly $d_{bh}$ is measured approximately at a height of 1.3 m. Assuming that $d_{bh}$ is a representative value of $b$, we can approximate the mass burning rate of porous wildland fire to be

$$q_{m,f} \approx 10^{-3} A_s d_{bh}^{-1/2}. \tag{B6}$$

Assuming that all of the trees from the ground to treetop are on fire, the exposed surface area of one tree is equal to $\pi d_{bh} h_t$, where $h_t$ is the average height of the burning trees. The exposed area of all the trees can therefore be approximated by

$$A_s = \pi d_{bh} h_t n_t A,$$
(B7)

where $n_t$ is the number of burning trunks per unit of forest area burning; and $A$ is the area of forest on fire.

The heat generated by a forest fire is estimated from Eqs. (1) and (B6). The lower heat of combustion ($H_c$) of woody fuel ranges typically from 17.8 to 20.4 MJ kg$^{-1}$ (e.g., Trentmann et al. 2006; Hurley, 2016).
We have therefore applied the middle value within this range ($H_c = 19.1$ MJ kg$^{-1}$).

The fraction of total energy released by combustion that is available for convection depends on the ambient and fuel conditions (Trentmann et al., 2006; Freitas et al., 2010; Kukkonen et al., 2014). Laboratory experiments of biomass burning (Freeborn et al., 2008) have indicated a mean convective
fraction of $51.8 \pm 9.0$ % (determined in terms of higher heat of combustion, i.e., including latent heat released during condensation of water vapour generated by the fire). We have assumed that 55 % of the total heat generated by a forest fire is available for convection ($\varepsilon_c = 0.55$). This is simply in the middle of the commonly accepted range from 0.4 to 0.8 (Trentmann et al., 2002; Freitas et al., 2010).

The emission factor can be defined to be the amount of the chemical species released per mass of dry biomass burned (e.g., Andreae and Merlet, 2001). Therefore, the emission factor is equal to the yield of combustion products ($y_i$). Data on emission factors for various types of biomasses burning have been presented by, for instance, Lemieux et al. (2004), Akagi et al. (2011), Kaiser et al. (2012) and Urbanski (2014a). The current model version applies the emission factors, which are applicable for the land cover
class of extratropical forest, presented by Kaiser et al. (2012). The extratropical forest class includes forest types typically found in the northern hemisphere (Kaiser et al., 2012).

For simplicity, particles formed in a forest or a liquid pool fire are assumed to be spherical. Further, they are assumed to be 2.5 μm in aerodynamical diameter having the density of water, i.e., the unit density = 1160    1 kg dm$^{-3}$.

**Appendix C. Centre line properties of a fire plume in the source term regime and the equivalent top-hat profiles**

The mean velocity ($u_0$) and excess temperature ($\Delta T_0$) at the centreline of a fire plume in the source term flow regime have been presented in Eqs. (8b) and (8c). These values approach their ambient values, as the radial distance from the plume centreline increases. We present in the following a model for determining the equivalent mean velocity and excess temperature for uniform (i.e., top-hat) profiles of the plume cross-sections, under the condition that convective heat energy is conserved.


We assume Gaussian radial profiles of the excess temperature, $\Delta T(r)$, and the mean velocity, $u(r)$ (Heskestad, 2016),

$$\Delta T(r) = \Delta T_0 \, exp\left(-\left(\frac{r}{\sigma_{\Delta T}}\right)^2\right), \tag{C1a}$$

$$u(r) = u_0 \, exp\left(-\left(\frac{r}{\sigma_u}\right)^2\right), \tag{C1b}$$

where $r$ is the radial distance measured from the centre line of the plume; and $\sigma_{\Delta T}$ and $\sigma_u$ are the measures of the plume width corresponding to the radial distributions of excess temperature and velocity, respectively. The density of the plume is assumed to have a constant value within each cross-section of the plume, equal to the centre line value ($\rho_0$).

The radius of the plume, $r_{\Delta T}$, has been defined in terms of $\Delta T_0$ (Eq. (8a)). A velocity radius ($r_u$) can be defined correspondingly: let $r_u$ be the plume radius at the point, at which the gas velocity has declined to $0.5 u_0$ (Heskestad, 2016). The temperature and velocity profiles have in general differing scales, i.e.,

$$r_u = a r_{\Delta T}. \tag{C2}$$

According to Heskestad (2016), an optimal value is $a = 1.1$, based on the most reliable measurements (George Jr. et al., 1977).

Applying Eqs. (C1a-b) and the definitions of the radiuses $r_{\Delta T}$ and $r_u$ yield an estimate for the measures of the plume widths,


$$\sigma_x = \left(ln(2)\right)^{-1/2} r_x \equiv b r_x,$$

(C3)

where the subscript $x$ is either $u$ or $\Delta T$; and $b \approx 1.201$.

The equivalent top-hat excess temperature ($\Delta T$) and velocity ($u$) of the plume can be derived by

integrating Eqs. (C1a) and (C1b), and using the relations (C3),

$$\Delta T = \Delta T_0 R^{-1} \int_0^R exp\left(-\left(\frac{r}{b r_{\Delta T}}\right)^2\right) dr,$$

(C4a)

$$u = u_0 R^{-1} \int_0^R exp\left(-\left(\frac{r}{b r_u}\right)^2\right) dr = u_0 R^{-1} \int_0^R exp\left(-\left(\frac{r}{a b r_{\Delta T}}\right)^2\right) dr,$$

(C4b)

where $R$ is a radial distance from the centre of the plume.

Equations (C4a-b) can be written more simply in terms of the error function ($erf$), defined as

$$\int_0^R exp\left(-\left(\frac{r}{\sigma}\right)^2\right) dr = \frac{\sqrt{\pi}}{2} \sigma \, erf\left(\frac{R}{\sigma}\right).$$

(C5)

Therefore

$$\Delta T = \Delta T_0 \frac{\sqrt{\pi}}{2} \frac{b r_{\Delta T}}{R} erf\left(\frac{R}{b r_{\Delta T}}\right) = \Delta T_0 \frac{\sqrt{\pi}}{2} \frac{b}{c} erf\left(\frac{c}{b}\right) \equiv \Delta T_0 s_{\Delta T},$$

(C6a)

$$u = u_0 \frac{\sqrt{\pi}}{2} \frac{a b r_{\Delta T}}{R} erf\left(\frac{R}{a b r_{\Delta T}}\right) = u_0 \frac{\sqrt{\pi}}{2} \frac{a b}{c} erf\left(\frac{c}{a b}\right) \equiv u_0 s_u,$$

(C6b)


where $c = R/r_{\Delta T}$, and $s_{\Delta T}$ and $s_u$ are dimensionless scale factors.

Substituting $\Delta T_0$ and $u_0$ in Eqs. (C1a-b) to Eqs. (C6a-b), and requiring conservation of convective heat flux yields


$$s_{\Delta T} s_u = \frac{\pi}{4} \frac{ab^2}{c^2} erf\left(\frac{c}{b}\right) erf\left(\frac{c}{ab}\right) = C^{-1}. \tag{C7}$$

Equation (C7) is an implicit function for $c$, which can be solved numerically. Let us next examine the properties of this numerical problem. From Eq. (C7) we may define a function $f$

$$f(x) = \frac{1}{x^2} erf(x) erf\left(\frac{x}{a}\right) - \frac{4}{\pi a C} \equiv \frac{1}{x^2} erf(x) erf\left(\frac{x}{a}\right) - D, \tag{C8}$$


where $x = c\, b\text{-}1 > 0$; and $D > 0$ is a constant. The function $f$ is continuous and differentiable. The zero point(s) of $f$ straightforwardly determines the radius $R$, and the scale factors $s_{\Delta T}$ and $s_u$.

Further, as $0 < erf(x) \leq 1$, for $x > 0$,


$$f(x) = \frac{1}{x^2} erf(x) erf\left(\frac{x}{a}\right) - D \leq \frac{1}{x^2} - D. \tag{C9}$$

Thus, $f(x) < 0$ for $x > D^{-1/2}$, and any possible zero points of $f$ are within $(0, D^{-1/2}]$. Applying the series expansion (e.g., Abramowitz and Stegun, 1972)

$$erf(x) = \frac{2}{\sqrt{\pi}} x \left(1 - \frac{x^2}{3} + \frac{x^4}{10} - \frac{x^6}{42} + \cdots\right), \tag{C10}$$


yields

$$\lim_{x \to 0} f(x) = \frac{4}{\pi a} - D = \frac{4}{\pi a}\left(1 - \frac{1}{C}\right). \tag{C11}$$

Hence, for


$$C > 1,\hspace{2cm}\text{(C12)}$$

$f$ is positive as $x \to 0$, and $f$ has at least one zero point. Function $f$, assuming typical values of experimental coefficients $a$, $C_1$, $C_2$ and $C_3$, has been illustrated in Fig. C1.

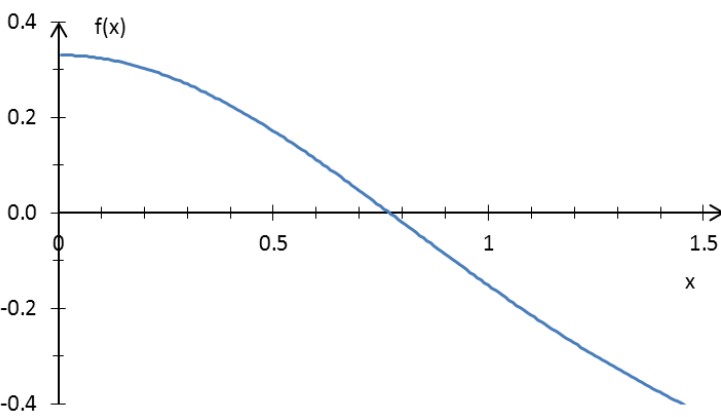


**Figure C1. Function $f(x)$ (defined in Eq. (B9)), with $a = 1.1$, $C_1 = 0.12$, $C_2 = 3.4$ and $C_3 = 9.1$.**

For $C < 1$, the conservation of convective heat energy cannot be achieved by applying the presented method. Therefore, any possible zero value of $f$ is physically irrelevant. The zero value of $f(x)$ was

estimated numerically $(x \in (0, D^{-1/2}])$ with a combination of linear interpolation, inverse quadratic interpolation, and bisection (Brent, 1971). Assuming that $C_1 = 0.12$, $C_2 = 3.4$, $C_3 = 9.1$ and $a = 1.1$, yields

$$c \approx 0.92578,\hspace{2cm}\text{(C13a)}$$
$$s_{\Delta T} \approx 0.83280,\hspace{2cm}\text{(C13b)}$$
$$s_u \approx 0.85788.\hspace{2cm}\text{(C13c)}$$

The value of $c$ (Eq. (C13a)) implies that the temperature and velocity profiles are integrated almost

(93 %) up to the point where the temperature excess has been declined to $0.5\Delta T_0$, whereas the velocity profile is integrated up to 84 % of $r_u$. The top-hat excess temperature ($\Delta T$) and velocity ($u$) are 83 % and 86 % of their centre line values, respectively, as written in Eqs. (12a-b).

Clearly, $s_{\Delta T} = s_u$ for distributions of $\Delta T(r)$ and $u(r)$ with equal scales ($a = 1$). Further, for $a = 1$, $s_{\Delta T} =$

$s_u = (\pi C_1^2 C_2 C_3)^{-1/2}$. For the same coefficients $C_i$ as above, yields $s_{\Delta T} = s_u \approx 0.84525$.

**Appendix D. An example application of the operational model.**

As an example of the model applications, we have simulated a forest fire that occurred on 25[th] July, 2021 in Kalajoki, western Finland. The burning area was approximately 70 hectares at maximum. The burning forest comprised mostly of trees of a height of approximately 16 m, the bole diameter of 20 cm and the density of trees 0.07 per square metre.

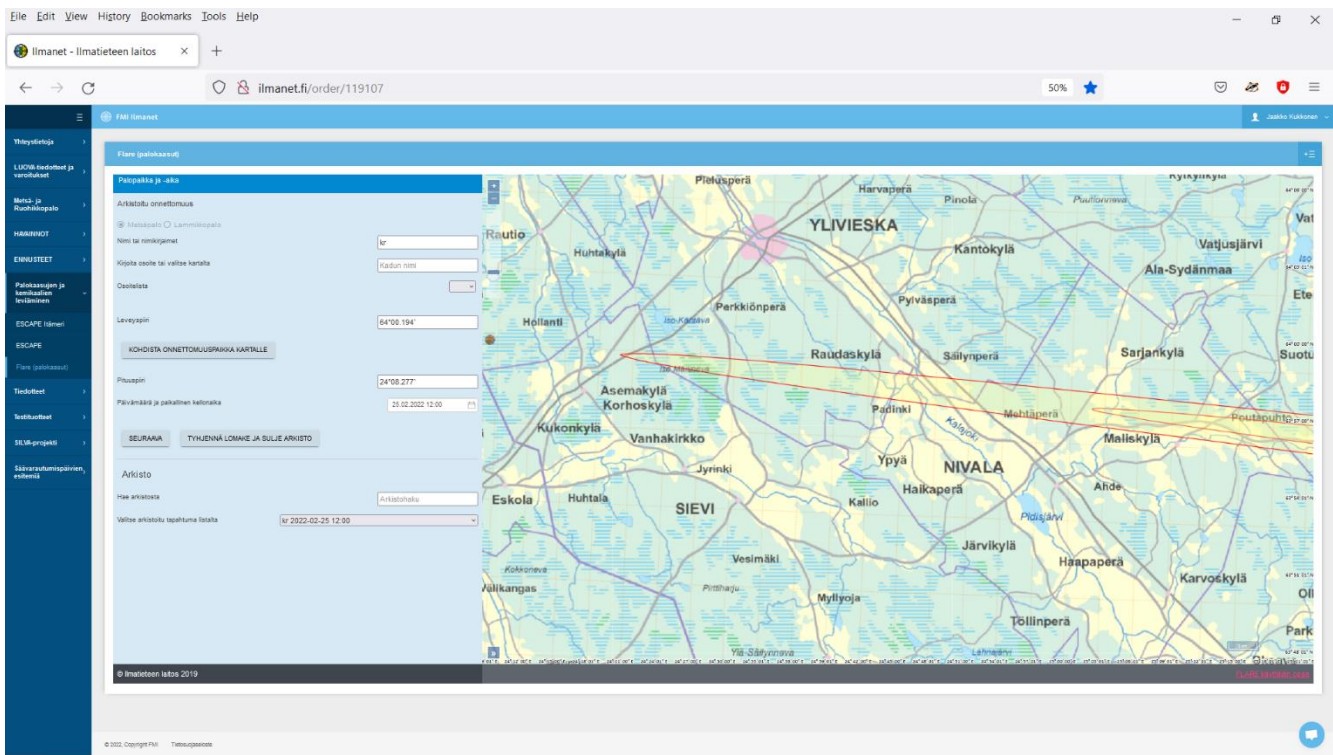

**Figure D1. A screenshot of an application of the operational model version. The plume was originating from a forest fire on 25[th] July, 2021 in western Finland. On the right-hand side, the concentrations of $CO_2$ at the ground level are presented on a map. The physical scale of the figure is approximately 20 km in the east-west direction. On the left-hand side, the model user has specified identification information for this case and has input the time and location of the fire on a map. The meteorological and forestry information have been automatically extracted for the computations.**

**Code and data availability**

The code and the relevant data are available in Zenodo at https://doi.org/10.5281/zenodo.4744300 (Kukkonen et al., 2021). These contain the source code of the BUOYANT model (v4.20), the technical reference of the model, the user manual of the model and the model input data corresponding to the work described in this paper. The model code, documentation and input data are published under the license of Creative Commons Attribution 4.0 International.


The experimental data of the RxCADRE campaign used in this paper can be downloaded from the Research Data Archive of the U.S. Department of Agriculture (Hudak et al., 2016a; Jimenez and Butler, 2016; Seto and Clements, 2015a and 2015b; Urbanski, 2014b).

**Author contribution**

The research version of the BUOYANT model, including the source term module, has been developed by Juha Nikmo, Jaakko Kukkonen and Kari Riikonen. All the authors have contributed to the development of the operational model version. Ilmo Westerholm, Pekko Ilvessalo, Tuomo Bergman and Klaus Haikarainen performed most of the research and coding that was necessary for the functioning of the operational model. Juha Nikmo performed the model computations for the evaluation of the model

against experimental data. Jaakko Kukkonen was the leader and coordinator of the project, in which this work was performed. Jaakko Kukkonen and Juha Nikmo prepared the manuscript, with contributions from all co-authors.

**Conflicting interests**

The authors declare that they have no conflict of interest.

**Acknowledgements**

We would like to thank for the financial support of the project "The assessment of the dispersion of pollutants from fires, for the needs of rescue services", funded by the Fire Protection Fund in Finland, and the project "Exposure to heat and air pollution in EUrope – cardio-pulmonary impacts and benefits

of mitigation and adaptation (EXHAUSTION)", Horizon 2020, European Union. Ministry of the Interior in Finland, especially Director General for Rescue Services Dr. Janne Koivukoski, is thanked for the continuous support of the project. We also wish to thank the following persons at the FMI for their contributions and comments in developing the operational model version: Mr. Juha Jääskeläinen, Mr. Ari-Juhani Punkka, Dr. Ari Karppinen, Dr. Antti Hellsten and Prof. Mikhail Sofiev. The authors also thank D.Sc. Matti Katila and his co-workers at Natural Resources Institute Finland for valuable advice regarding the use of the Multi-Source National Forest Inventory of Finland.

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
