# Peer review of "An emergency response model for evaluating the formation and dispersion of plumes originating from major fires (BUOYANT v4.20)"

_Geoscientific Model Development, 2021_

## Referee Comment (RC1)

Reviewer Report:
**An emergency response model for evaluating the formation and dispersion of plumes originating from major fires (BUOYANT v4.20)**
* * *
The paper presents recent developments implemented within an existing plume dispersion model for forest and pool fires, which aim to improve how the emissions source is parameterized within the modelling system. Given the sensitivity of plume rise and subsequent dispersion to source input parameters, estimation of buoyancy, mass and momentum fluxes is key to improving model prediction accuracy. Unlorturnaly, while this paper provides the means to estimate these parameters, it is missing key model evaluation to support the presented approach.

**General Comments**

Moving away from static inputs towards a more physical model for fire source parameters is incredibly valuable. The authors present an approach for estimating various source properties derived from the classic MTT model, which in my view constitutes the main contribution of the paper. However, no attempt is made to actually evaluate this "source term model".

The inter-comparison study with RxCADRE data presented in the paper specifically excludes the source term model, focusing instead on the two previously-studied components of BUOYANT (plume rise and dispersion). While such results are still valuable, they do not substitute for proper evaluation of the derivations presented in Section 2 and, in the current state, provide no supporting evidence for the main contribution of the paper.

Fortunately, RxCADRE dataset is incredibly detailed and can be used to extend the evaluation to include the source term model.  My recommendation for the publication of this paper would, hence, be contingent on the authors demonstrating the results for the following:

Comparison of RxCADRE observations to:
        - BUOYANT model with ***observations*** as source inputs (this is essentially what's currently included in the paper), with more details included on methodology (as per comments below)
        - BUOYANT model with ***old*** source term parameterization (fixed parameters)
        - BUOYANT model with ***new*** source term model included
        - Operational version of BUOYANT (if different from above)

Lastly, Section 4 of the paper is dedicated entirely to an overview of an operational modelling system. It is my understanding that the system is supposed to be accessible online, however no links are provided in the paper (aside from those pointing to an offline archive of the Fortran source code for the BUOYANT model). My current review of Section 4 is, hence, fairly superficial. If the authors are unable to provide access to the model for peer-evaluation, my recommendation would be to exclude this section from the manuscript.

**Specific Comments**

*Section 1*
The current overview is rather scattered, jumping from plume rise, to combustion, to air quality modeling , to emissions, to CFD models and, finally, to BUOYANT, with little effort to connect the topics. The broad context should be covered in a more concise and logical way (e.g. are satellite emissions studies and CFD models even relevant?), while the work pertaining directly to BUOYANT model needs to be covered in greater detail.

Line 34: Please include more recent review works on plume rise from wildfires
Line 35: How does this section connect to the previous one? How are combustion studies relevant?
Line 61: "satellite-based estimates of wildland fires": what does that mean? Emissions from wildland fires?
Line 63, 65: why should measuring FRP "require inputs"?  How are FRP-based emissions estimates relevant to this work?
Line 71: how does this section connect to the previous one and what is the relevance?
Line 80: Given the overview of literature provided, what are the current knowledge gaps this manuscript is going to address? What is missing?
Line 81-87: please include more detail about literature on the BUOYANT model and the relevant findings
Line 93: what do authors mean by "all other models"?

*Section 2*
Figure 1(b). Does the model actually include full vertical profiles, or is it a two-layer atmosphere? I.e. please move all information from Appendix D into the body of the paper. It's critical for understanding how the model works.
Line 139 - 150: This belongs in Introduction
Line 148: Which "model"? What were the specific findings? If the agreement was great, why would BUOYANT need improvement? What were the limitations?
Line 151: What do authors mean by "treatments"? Have the above-mentioned model evaluation studies all relied on these three fixed parameters? What are the parameters?
Line 174: Which "separate models" are the authors referring to?
Line 175: I struggle with this assumption for wildland fires. Winds modify convective fluxes, also wildfires generate their own winds. Please provide more support for this. Thorough model evaluation with observational data would, of course, constitute the best supporting evidence.
Line 200: How does the model handle fuel moisture then? It's a critical parameter for wildland fires.
Line 224: So which terms are going to be used in the subsequent modelling steps? A summary table would be helpful.
Line 257: Please clarify further what atmospheric stability structure is considered. Does "calm" refer to neutral stably stratified vertical profile?
Line 258: MTT applies specifically to point sources
Line 280: The original entrainment assumption of the MTT model doesn't hold for wildland fires. Has the RS entrainment model been used for such scenarios? If so, please provide a reference. If not, please provide support for the assumption.

Line 316: How does this connect to the previous paragraph? As mentioned above, a summary table of modeled parameters can likely help with clarity and structure.

Eqn 10: At which height? I.e. this is not considering entrainment/detrainment. Please clarify the assumptions used for your definitions

Line 348: Which "model" and which "computer model". Please see my suggestion in the Writing Style section.

*Section 3*

As noted above, my main concern with this model evaluation is that it ignores entirely what has been presented in the previous parts of the paper. Moreover, evaluation of the plume rise and dispersion components of BUOYANT has been previously performed, based on authors' own Introduction. What is then the main goal and novelty of this section then? It is critical that this portion of the manuscript is expanded to include a full evaluation of the source term module (see General Comments above).

Please make a table of all required model input parameters, showing the estimates based on (a) RxCADRE data (b) the source term model presented. Please include a description of how each value was obtained, pointing to specific equations, data sources etc.

Lastly, please include a detailed discussion (either under model evaluation, or as a separate section) addressing model performance and placing it in context of earlier published work.

Line: 406: Please include the following information: dimensions of the burn, how it was inited (and how long was the ignition process), duration of the active burning.

Line 420: Which heat fluxes, specifically? How was FRP estimated?

Line 450: There are vertical atmospheric soundings that were collected during the experimental campaign (see Clements 2016). Are these used for the study? Or does it solely rely on a 30m tower observations? If so, please explain why.

Line 456: Please create a Figure showing the observed vs. modeled atmospheric profile.

Line 458: Were the winds considered to be uniform? Is BUOYANT able to account for vertical wind shear? Please move all the answers to these questions out of Appendix D and into the paper.

Line 466: The paper presents a new source term model. Model evaluation, therefore, CANNOT exclude the source term parameterization.

Line 471: "the BUOYANT model" as per Line 121, *includes* source term module. As noted above, please provide the estimates of all model input parameters using the source term module.

Line 484: Please plot the used FRP and area values on Figure 2 as a horizontal lines. They seem a bit low to represent the average (though my eye-balling could be wrong). It would be great, if the authors presented some sort of sensitivity analysis of the model to these input parameters.

Line 486: Abstract states that the BUOYANT model estimates cross-plume integrated values. My understanding is that in the current section the authors are presenting instantaneous concentrations. Please explain the discrepancy.

Line 507: "temporally well-captured" is a bit generous. Figure 3(a) estimates plume rise to be at half the measured value.

Line 513: Are these middle points in 3(b) calculated using center of mass? (they probably should be).

Line 519: There's quite a bit of a disagreement, actually (for good reasons). 4(a) shows no smoke above 590m, while observed values extend to 1230m. There's likely directional wind shear present, as higher observed values are shifted in the cross-wind directions. As noted above, a plot of vertical vs modeled atmospheric profile is much needed. Figure 3(b) drastically underestimates plume width. Once again, there's likely a logical explanation for this. Namely, as far as I recall, the lot was ignited with multiple straight lines (strip head fire ignition) of roughly 1km length, and the model attempts to represent the same fire strips with a circle of 158m radius (based on the configuration file provided in supplementary material). One shouldn't expect this assumption to work well in the given scenario. Figure 3(c) again, suggests lack of wind shear and strong concentration overestimation. Please include an objective description of model performance here, as well as in Discussion. Also, please explain why the observed values at 280m (green line) in 3(c) simply cut off. Lastly, there appears be no plume growth and widening at all: it remains 2km wide from 1.7km to 11km downwind. Please explain. This, in part, explains the overestimation of concentration values further downwind.

Line 528: Please be more specific: what are the differences? What are the challenges? Wasn't the source term model designed to address these challenges?

Line 541: What meteorological inputs are referred to, specifically? Full vertical atmospheric profile was measured during the RxCADRE campaign, so the vertical structure did not need to be estimated. Again, please include a figure for the vertical profile used in BUOYANT vs observations.

Line 545: Given observed values were available, this argument cannot be used to explain the discrepancies between modelled and measured concentrations.

*Section 4.*

Please provide a link to the online operational version of the model (if it's indeed available in the paper somewhere and I missed it, my apologies! Please move the link the beginning of Section 4.1 for easy access).

Without access to the system, the following comments do not constitute proper peer-review. Please also include the operational version of the model in the evaluation.

Line 632: Ambient pressure, wind and temperature profiles are standard model output. What's being "evaluated"?

Line 635: Is roughness length the only parameter that actually needs estimation?

Line 660: What do authors mean by "efficient functioning"?

*Conclusions:*

Here the authors seem to mix summary with discussion. Please see my earlier comment, regarding clearly separating out results from discussion.

Line 665: As above, please clarify whether this is a cross-plume integrated model or not. Figure 4 would only be possible if the fields are NOT cross-plume integrated.

Line 671: which characteristics, specifically?

Line 684-687: Description of the campaign belongs in the intro/methods, not in conclusion.

Line 687-691: Please see comments to Section 3.

Lines 693-712: This very much resembles a Discussion section, and should be relocated away from Conclusions, accordingly, together with responses to comments for Line 519 (please see above)

Line 696: What are the challenges, specifically? What are the major sources of uncertainty, specifically?

*Appendix C*
Line 935: that seems to be a fairly extreme assumption, I am very curious to see how the source term model performs for the L2F burn.

*Appendix D*
Please move this into the main body of the paper.

**Writing Style**
The authors use the term "evaluate" extensively, to refer to both "estimation" and "measurement" of a particular parameter. My recommendation would be to specifically indicate whether a particular term is *estimated* (i.e. modelled) vs. *measured* (i.e. obtained from observations) vs. *calculated.*

The variable use of term "model" to refer to BUOYANT with multiple components, as well as to BUOYANT without the source term model, to the source model and to the computer model is rather confusing. My suggestion would be to call BUOYANT (with all three components) a model, and individual sub-models (dispersion, plume rise, source term) - modules. Operational version can be referred to as BUOYANTops, or something on those lines.

Please note that the draft requires extensive English language editing.

**References**

Clements Craig B., Lareau Neil P., Seto Daisuke, Contezac Jonathan, Davis Braniff, Teske Casey, Zajkowski Thomas J., Hudak Andrew T., Bright Benjamin C., Dickinson Matthew B., Butler Bret W., Jimenez Daniel, Hiers J. Kevin (2015) Fire weather conditions and fire–atmosphere interactions observed during low-intensity prescribed fires – RxCADRE 2012. *International Journal of Wildland Fire* **25**, 90-101.

---

## Author Comment (AC1)

**Authors' preliminary comments to Referee no 1**

The comments of the referee have been presented in *blue italic* below, and our response as plain black text. We have considered at this point only major comments of the reviewer, not the smaller specific comments.

First, we would like to thank reviewer no. 1 for the pertinent, useful and detailed comments on our manuscript. We would be very happy to write a revised version taking into account his/her comments.

*The paper presents recent developments implemented within an existing plume dispersion model for forest and pool fires, which aim to improve how the emissions source is parameterized within the modelling system. Given the sensitivity of plume rise and subsequent dispersion to source input parameters, estimation of buoyancy, mass and momentum fluxes is key to improving model prediction accuracy. Unlorturnaly, while this paper provides the means to estimate these parameters, it is missing key model evaluation to support the presented approach.*

The main aim of developing the source term module for the buoyant plume dispersion model was actually **to make the use of the overall model much easier**. The required input data for the refined model is indeed substantially simpler and easier to estimate, compared with those of the original model (i.e., the model without the source term module). **The main aim was not to obtain more accurate predictions** with the revised modelling system. This should be stated much more clearly in the revised manuscript.

However, we acknowledge the importance of the reviewer's statement. We could therefore inter-compare the results obtained using the source term model and those obtained using the original model (without the source term module), using RxCADRE data, and include these comparisons to the revised manuscript. We discuss this in more detail below.

Unfortunately, at the time of writing the manuscript we did not find any suitable datasets in the publicly available literature for evaluating separately the source term model directly against

experimental data. We therefore did what is possible in practise; i.e., we have evaluated the whole model (although not specifically the source term module) against the best available dataset.

Generally, it is fairly common in the literature that regarding large and complex models, not each and every module included in the overall modelling system will be separately tested. There might be several reasons for this, one of which is simply the lack of sufficient quality experimental data on that specific aspect of the model. In some cases, one has to satisfy to simply evaluate the whole model, including all the separate modules – although this process of course may not critically test all the modules included.

**General Comments**

*Moving away from static inputs towards a more physical model for fire source parameters is incredibly valuable. The authors present an approach for estimating various source properties derived from the classic MTT model, which in my view constitutes the main contribution of the paper. However, no attempt is made to actually evaluate this "source term model".*

*The inter-comparison study with RxCADRE data presented in the paper specifically excludes the source term model, focusing instead on the two previously-studied components of BUOYANT (plume rise and dispersion).*

This is correct. It should also be stated more clearly in a revised manuscript.

*While such results are still valuable, they do not substitute for proper evaluation of the derivations presented in Section 2 and, in the current state, provide no supporting evidence for the main contribution of the paper.*

*Fortunately, RxCADRE dataset is incredibly detailed and can be used to extend the evaluation to include the source term model. My recommendation for the publication of this paper would, hence, be contingent on the authors demonstrating the results for the following:*

We totally agree with the reviewer that RxCADRE is a useful dataset. It has also been well documented. We have re-checked the data and conclude that we could make some intercomparisons of the predictions obtained using the source term model and the original (generic input data) model. We would like to suggest that such comparisons would be included and discussed in the revised manuscript.

*Comparison of RxCADRE observations to:*
*BUOYANT model with **observations** as source inputs (this is essentially what's currently included in the paper), with more details included on methodology (as per comments below)*

This is a reasonable request and can be done.

*BUOYANT model with **old** source term parameterization (fixed parameters)*
*BUOYANT model with **new** source term model included*
*Operational version of BUOYANT (if different from above)*

The BUOYANT model makes it possible to evaluate strongly buoyant plumes in two alternative ways: (i) Using the source term model, as presented in the manuscript, or (ii) describing the source parameters to the overall buoyant plume model (without using the source term model). In the latter case, the input data is much more complex and much more difficult to evaluate. These two options could be compared for the RxCADRE data, as the reviewer suggests. However, at least in case of forest fires, experimental data for performing such an inter-comparison is not currently available in the available literature, except.

Both the operational version of the model (named as FLARE) and the original research model (named as BUOYANT) use an identical code for the dispersion and transport of a buoyant plume. The differences of FLARE and BUOYANT are:

(a) FLARE uses as default the presented source term model; it does not therefore allow the user to specify the source related input in the more complex format (as in the alternative (ii) in the above paragraph).

(b) The specification of the meteorological conditions in FLARE is determined by the used numerical weather prediction model, in BUOYANT, this can be done also in various other ways.

(c) Output of BUOYANT is more versatile and can be adjusted by the user, compared to FLARE.

(d) BUOYANT allows the user to post-process the model results at will. FLARE includes a standard format post-processing.

(e) FLARE can be used with a restricted set of web browsers, while BUOYANT is designed to be as platform independent as possible.

These differences should probably be presented more clearly in a revised manuscript.

*Lastly, Section 4 of the paper is dedicated entirely to an overview of an operational modelling system. It is my understanding that the system is supposed to be accessible online, however no links are provided in the paper (aside from those pointing to an offline archive of the Fortran source code for the BUOYANT model). My current review of Section 4 is, hence, fairly superficial. If the authors are unable to provide access to the model for peer-evaluation, my recommendation would be to exclude this section from the manuscript.*

We apologise for not specifying a link for the operational model version; this was an omission. We can and should provide the link for the reviewers. A slight problem is that the FLARE user interface has been currently coded only in Finnish. However, we suggest to provide the translations of the relevant texts to English as a separate document. There is only one or a couple of pages of text in this user interface. We therefore believe that this would give the reviewers a sufficient knowledge on the functioning of the software.

*… If the agreement was great, why would BUOYANT need improvement? What were the limitations? (specific comment)*

The limitations of the BUOYANT model include:  (i) The model assumes a steady state in terms of emissions and meteorology. However, the user can easily conduct multiple runs with various values of the emissions and meteorological parameters, to evaluate the impacts of changing emission and ambient conditions. (ii) The current model version does not treat the impacts of phase changes of water in the plume (in particular, condensation and evaporation). (iii) The model adopts some values of model parameters according to the best available previous experimental and modelling studies. However, the values of these parameters could be found to be inaccurate in the future and may have to be refined.

---

## Author Comment (AC2)

**Authors' comments to Referees no 1 and 2**

The comments of the referee have been presented in blue font below, and our response as plain black text. We have considered at this point mainly the major comments of the reviewers.

**Authors' comments to Referee no 1**

The paper presents recent developments implemented within an existing plume dispersion model for forest and pool fires, which aim to improve how the emissions source is parameterized within the modelling system. Given the sensitivity of plume rise and subsequent dispersion to source input parameters, estimation of buoyancy, mass and momentum fluxes is key to improving model prediction accuracy. Unlorturnaly, while this paper provides the means to estimate these parameters, it is missing key model evaluation to support the presented approach.

**General Comments**

Moving away from static inputs towards a more physical model for fire source parameters is incredibly valuable. The authors present an approach for estimating various source properties derived from the classic MTT model, which in my view constitutes the main contribution of the paper. However, no attempt is made to actually evaluate this "source term model".

The inter-comparison study with RxCADRE data presented in the paper specifically excludes the source term model, focusing instead on the two previously-studied components of BUOYANT (plume rise and dispersion).

While such results are still valuable, they do not substitute for proper evaluation of the derivations presented in Section 2 and, in the current state, provide no supporting evidence for the main contribution of the paper.

Fortunately, RxCADRE dataset is incredibly detailed and can be used to extend the evaluation to include the source term model. My recommendation for the publication of this paper would, hence, be contingent on the authors demonstrating the results for the following:

Comparison of RxCADRE observations to:

BUOYANT model with **observations** as source inputs (this is essentially what's currently included in the paper), with more details included on methodology (as per comments below)

BUOYANT model with **old** source term parameterization (fixed parameters)

BUOYANT model with **new** source term model included

Operational version of BUOYANT (if different from above)

We totally agree that it is unfortunate that the source term module could not be convincingly evaluated using the RxCADRE data. The reasons for this should be discussed in the revised manuscript. The main reason is that in the RxCADRE experiment, the actual trunks of the trees were not burnt; whereas the operational BUOYANT model assumes that this is the case.

According to the referee's advice, we have nevertheless also done a comparison of the source term model predictions with the RxCADRE data and presented those together with the results obtained using the original model (using measured input data, without the source term module). We suggest including also this comparison to the revised manuscript. This comparison could be useful to illustrate the potential use of the operational model in cases, in which the forest fire does not burn the actual main tree structures, i.e., the trunks of the trees.

We have also conducted a sensitivity study related to the model evaluation; we propose that this will be included to the revised manuscript. The sensitivity study evaluates the impact of the changes of input data to the predictions, especially regarding the influence of such input data quantities, the values of which were uncertain in the original RxCADRE data archive.

Unfortunately, at the time of writing the manuscript we did not find any suitable datasets in the publicly available literature for evaluating separately the source term model directly against experimental data (including burnt tree trunks). We therefore did what is possible in practise; i.e., we evaluated the whole model (although not specifically the source term module) against the best available dataset.

We also would like to mention that the main aim of developing the source term module for the buoyant plume dispersion model was actually to make the use of the overall model easier for the rescue personnel. The required input data for the refined model is indeed substantially simpler and easier to estimate, compared with those of the original model (i.e., the model without the source term module). The main aim was not to obtain more accurate predictions for all the conceivable

forest fire cases with the revised modelling system. In major forest fires, which are the most important for rescue operations, also the tree trunks, or at least part of the tree trunks will be burnt. However, in all openly available prescribed burning forest fire experiments, that is unfortunately not the case. We suggest to describe these arguments more clearly in the revised manuscript.

Regarding the evaluation against data of large and complex models, it is fairly common that not exactly each and every module included in the overall modelling system will be separately tested. There might be several reasons for this, one of which is simply the lack of sufficient quality experimental data on that specific aspect of the model. In some cases, one has to satisfy to simply evaluate the whole model, including all the separate modules – although this process of course may not critically test all the modules included.

We totally agree with the reviewer that RxCADRE is a useful dataset. It has also been well documented.

Lastly, Section 4 of the paper is dedicated entirely to an overview of an operational modelling system. It is my understanding that the system is supposed to be accessible online, however no links are provided in the paper (aside from those pointing to an offline archive of the Fortran source code for the BUOYANT model). My current review of Section 4 is, hence, fairly superficial. If the authors are unable to provide access to the model for peer-evaluation, my recommendation would be to exclude this section from the manuscript.

We apologise for not specifying a link for the operational model version. The link is http://ilmanet.fi/order/147135, user name: GMD, password: FLARE21!. A slight problem is that the FLARE user interface has been currently coded only in Finnish. However, we will provide the translations of the relevant texts to English as a separate document. We believe that this would give the reviewers a sufficient picture on the functioning of the software.

Short user guide in English (attachment).

Page 1: Login, user name: GMD, password: FLARE21!

Page 2: General information concerning the location of the burning and time of forest or pool fire. E.g., select 'Forest fire'.

Page 3: Editing of weather and forest parameters. Weather parameters are produced automatically by the by NWP model HARMONIE.

Page 4: Same as 'Page 3', but now 'Extended weather parameters' selected.

Page 5: 'Pool fire' option selected.

… If the agreement was great, why would BUOYANT need improvement? What were the limitations? (specific comment)

The main aim of developing the source term module for the buoyant plume dispersion model was practical, to make the use of the overall model easier for the rescue personnel. This has been elaborated above.

The limitations of the BUOYANT model include: (i) The model assumes a steady state in terms of emissions and meteorology. However, the user can easily conduct multiple runs with various values of the emissions and meteorological parameters, to evaluate the impacts of changing emission and ambient conditions. (ii) The current model version does not treat the impacts of phase changes of water in the plume (in particular, condensation and evaporation). (iii) The model adopts some values of model parameters according to the best available previous experimental and modelling studies. (iv) The chemical reactions of pollutants during the source term and plume rise stages are not addressed.

We suggest that these limitations will be better described in a revised manuscript.

**Authors' preliminary comments to Referee no 2**

General Comment:
The paper presents the development of a source term model that evaluates the fire plume properties just above the flame as an extension to the previously published BUOYANT model for the dispersion of buoyant plumes from wildfires and liquid pool fires under varying atmospheric conditions. The refined BUOYANT model v4.20 is then evaluated against observational data of $CO_2$ concentrations from aircraft measurements during a wildfire experiment from the RxCADRE

campaign. The model captured well the vertical profiles of $CO_2$, while the highest concentrations were moderately overpredicted. The authors also state that the widths of the plumes are slightly underestimated, without giving an explanation for this behavior.

We will expand the discussion on the differences of model predictions and measured data in the revised manuscript, including especially the widths of the plumes.

Further, an operational version of the BUOYANT model, called FLARE, is briefly presented, although it remains somewhat unclear how it is related to the research version of the model.

This is a good comment and deserves clarification in the manuscript. These model versions are closely connected with each other. Both the operational version of the model (named as FLARE) and the original research model (named as BUOYANT) use an identical code for the dispersion and transport of a buoyant plume. The operational model includes the research code in full, and all the core physics computations in the operational model are done using the research code. However, there is a dedicated user interface in the operational model, to facilitate an easier operational use. These connections will be explained in more detail in the revised manuscript.

In more detail, the differences of FLARE and BUOYANT are:

(a) FLARE uses as default the presented source term model; it does not therefore allow the user to specify the source related input in the more complex format (as in the alternative (ii) in the above paragraph).

(b) The specification of the meteorological conditions in FLARE is determined by the used numerical weather prediction model, in BUOYANT, this can be done also in various other ways.

(c) Output of BUOYANT is more versatile and can be adjusted by the user, compared to FLARE.

(d) BUOYANT allows the user to post-process the model results at will. FLARE includes a standard format post-processing.

(e) FLARE can be used with a restricted set of web browsers, while BUOYANT is designed to be as platform independent as possible.

These differences should be presented more clearly in a revised manuscript.

Currently, the evaluation against the experimental data from the wildfire plume is performed with the refined BUOYANT model, but not including the extension with the new source term model. It is a bit unfortunate that the validity of this essential new module has not been demonstrated. My recommendation is to carry out a dedicated sensitivity study of the possible input value ranges in the source term model and after interfacing it with the BUOYANT model, comparing to the experimental data.

We totally agree that it is unfortunate that the source term module could not be convincingly evaluated using the RxCADRE data. The reasons for this will be discussed in the revised manuscript. The main reason is that in the RxCADRE experiment, the actual trunks of the trees were not burnt; whereas the operational BUOYANT model assumes that this is the case.

However, the reviewer's suggestion of a sensitivity study is very useful. We have already conducted such a sensitivity study regarding the impact of the changes of input data to the predictions. Such results are useful and illustrative, especially regarding the influence of those input data quantities, the values of which were uncertain in the original RxCADRE data archive.

We have also included a comparison of the source term model predictions with the RxCADRE data. Although the amount of wood burnt cannot be accurately evaluated with the new source term model in that case, this comparison could be useful to illustrate the potential use of the operational model in cases, in which the forest fire does not burn the actual main tree structures, i.e., the trunks of the trees.

Nevertheless, I think that the development of a physical model that can deal with the early evolution of the fire plume, the plume rise, and the local atmospheric dispersion is of great value for achieving advances in the prediction of impacts from major natural and man-made fires. Overall, the paper deserves publication after my general comment and the specific comments below are sufficiently addressed.

Thank you for the pertinent comments.

Specific Comments:

1.) P. 2, lines 48-51: It is mentioned that hundreds of chemical compounds are emitted into the atmosphere during wildland fires. Table C1 also lists a number of different fuels for which the convective heat flux and mass fluxes during a liquid pool fire can be calculated. How much of the combustion chemistry and oxidation is BOUYANT capable of simulating?

The mass burning rates (i.e., rate of mass burned per time) from a liquid pool or forest fire are estimated with semi-empirical correlations (as described in detail in Appendix C of the manuscript). Estimates of substance-specific fire products (e.g. mass flux of $CO_2$) are obtained from the modelled mass burning rate and experimentally derived emission factors. Currently applied substance-specific emission factors for liquid pool fires are shown in Table C1 of the manuscript. For forest fires we have applied the substance-specific emission factors presented by Kaiser et al. (2012).

However, after the emissions have been modelled for a range of chemical compounds, the model does not address chemical reactions within the source term and the plume rise regimes. The model can of course predict the plume properties after those regimes and then be coupled with dispersion models that address chemistry. For instance, the BUOYANT model has previously been used in that way in combination with the chemical transport model SILAM (which contains a chemistry sub-model including more than hundred chemical compounds).

As far as we know, none of the publicly available plume rise models treat explicitly the chemical reactions during the plume rise regime.

2.) I suggest to revise the paragraph on CFD models in the Introduction (P. 3, lines 70-80) to address the treatment of plumes from the two different types of fires: wildfires and liquid pool fires. Currently, only the dispersion models for treating liquid pool fires are described. Further, the respective description should deal with above-fire (source term), plume rise and large-scale dispersion for the two fire types.

We will revise this paragraph to be more clear and to address also forest fires. The CFD models have been used for estimating liquid pools fires, as described in the manuscript. However, we are not aware of studies, in which these would have been used for analyzing forest fires. However, the physics of such models is the same irrespective of the source of the buoyant plume. That is the

reason for using only one computer module for analyzing the plume rise regime (after the source term regime), also in the BUOYANT model. The source term model evidently has to be different for the forest fire and pool fire cases.

At which point, i.e. distance from source and vertical layer, does a large scale atmospheric model take over?

The criteria for the termination of the plume rise stage have been described in detail in our previous article, Kukkonen et al. (2014) (section "2.1.6 Criterion for the termination of plume rise"). The criterion used in the BUOYANT model is the following: "In the current model version, we have chosen simply to use the height, at which buoyancy force first vanishes as the final rise height of the plume." In short, when there is no buoyancy left in the plume, the model makes the transition from the plume rise regime to the dispersion modelling regime.

This distance (and the corresponding height) clearly depends a lot on the intensity and area of the fire, and on the meteorological conditions. In the RxCADRE model simulation presented in the article the plume rise stage comes to an end at the distance of 6.2 km from the source, at a height of 510 m (the height at the plume centerline).

3.) Section 2.2: it is not really clear from the descriptions in this section, how the new source term model for evaluating the properties of the fire plume above the flame tips is interfaced with the plume dispersion in the BUOYANT model.

This is a good point. This transition has been described in the section 2.2.4 (from the source term to the plume rise regime). However, the description is probably too brief and general.  We will add a more specific description to that section. This has actually been described in our previous article Kukkonen et al. (2014), quote: "Information on the source term (to be input to the plume rise module) includes the following: the source radius, the source height above the ground, the temperature of the released mixture of contaminant gas and particles, and air, the mass flux of this mixture, the mass fraction of the released gas, and the molecular weight and heat capacity of the released gas.".

4.) Section 3.3: as it is now, the evaluation is done for the BOUYANT model without the source term model. This may appear reasonable at first since the fire properties for the selected wildfire case L2F are well characterized by the observations. However, in order to evaluate the source term model presented in this paper, it would make sense to conduct a sensitivity analysis, studying the probable value ranges of selected fire parameters, and comparing the outputs of the BOUYANT model - including the source term sub-model - against the measurements in L2F.

Yes, we agree. Our plans to remedy this have been described above.

5.) Section 3.3.1: which method was used to evaluate the meteorological parameters on P. 18, lines 458-464, for use in the BOUYANT model?

The methods to evaluate the Monin-Obukhov length, and the vertical profiles of wind speed and temperature are briefly described in the second paragraph of the section 3.3.1. We suggest to elaborate these in a revised manuscript.

Prevailing wind direction was assumed to be the arithmetic mean of the experimental wind directions at the heights 6.2 m and 990 m. Atmospheric pressure was determined utilizing the measured ambient temperature profile and the hydrostatic assumption. The measured vertical temperature profile indicates (by visual inspection) a boundary layer height of approximately 2.2 km.

6.) Section 3.3.2: a table should be provided that contains the required input parameters and the values used in the evaluation both for the BUOYANT model without the source term model (currently listed on P. 19, lines 483-484) and for the BOUYANT model when using the source term model.

This is a good idea, and will clarify the model application. We have compiled tables containing the input parameters of the model, as used with and without the source term model. We suggest to add these to the manuscript.

7.) P. 23, line 536: the temporal evaluation should also be shown in a plot, for example at 450 m height above ground at the plume centerline.

This is a reasonable suggestion. However, we suggest to remove the sentence on line 535: "However, the temporal agreement of the measured and modelled highest concentrations was good." The aircraft measurements represent centered moving average over 20 seconds, at certain average three-dimensional locations. The exactly corresponding spatial and temporal averages were computed with the model and compared with the measured values. Although both of these have been carefully performed and archived, it is therefore in our view an over-statement to write something about the temporal evolution of measurements vs. data (as both the time and location are changing simultaneously). However, we can compare the measured data and predictions for certain flight manoeuvres, such as the crosswind distributions in Fig. 4.), but not really versus the time.

8.) Section 4: I think the presentation of the operational version of the model should be placed before the evaluation chapter 3 to make it more visible to the readers.

Yes, this can be done.

The operational version FLARE needs to be better related to the research version of the model. Does it use the new source term module?

This has been described above in response to the reviewer's general comments.

Information should be added about the stakeholder groups that are targeted as potential users. It would be nice to include a screenshot from a real-world example application.

Yes, these will be added to the manuscript. The stakeholders and users of the operational model include currently a wide range of emergency response personnel in Finland, the operational meteorologists at the Finnish Meteorological Institute (FMI), the Ministry of the Interior in Finland and researchers at the FMI. This user group could be expanded to other countries in the future.

9.) P.28, lines 689-690: statement "for most of the highest concentrations" seems to contradict with the finding of moderate overprediction of the highest concentrations.

Yes, quote: "the model captured well the observed vertical excess concentration distributions" is an overstatement and will be revised. The model predicted the heights for the "parking garage" (PG) measurements, there were 11 measured or predicted peak values. Six of these were over-predictions and 5 were under-predictions by the model. In addition, the model predicted one peak that was not found in the measurements. The model under-predicted the measured data in PG#1:n and over-predicted those in PG#2 and PG#3.

The sensitivity analysis that we performed after submitting the manuscript actually improved our understanding of the reasons for these deviations of data and predictions.

Technical Corrections
Figure 4, middle plot: the green curves from modelling and measurements do not show well.

We have corrected Figure 4b.

**References**

Kaiser, J.W., Heil, A., Andreae, M.O., Benedetti, A., Chubarova, N., Jones, L., Morcrette, J.-J., Razinger, M., Schultz, M.G., Suttie, M., and van der Werf, G.R.: Biomass burning emissions estimated with a global fire assimilation system based on observed fire radiative power, Biogeosciences, 9, 527-554, doi:https://doi.org/10.5194/bg-9-527-2012, 2012.

Kukkonen, J., Nikmo, J., Sofiev, M., Riikonen, K., Petäjä, T., Virkkula, A., Levula, J., Schobesberger, S., and Webber, D.M.: Applicability of an integrated plume rise model for the dispersion from wild-land fires, Geosci. Model Dev., 7, 2663-2681, doi:https://doi.org/10.5194/gmd-7-2663-2014, 2014.

---

## Author Response (AR1)

**Authors' comments to Referees no 1 and 2**

The comments of the referee have been presented in blue font below, and the authors' responses in plain black text. Revisions done to the manuscript have been briefly explained in red font.

**Authors' comments to Referee no 1**

The paper presents recent developments implemented within an existing plume dispersion model for forest and pool fires, which aim to improve how the emissions source is parameterized within the modelling system. Given the sensitivity of plume rise and subsequent dispersion to source input parameters, estimation of buoyancy, mass and momentum fluxes is key to improving model prediction accuracy. Unlorturnaly, while this paper provides the means to estimate these parameters, it is missing key model evaluation to support the presented approach.

**General Comments**

Moving away from static inputs towards a more physical model for fire source parameters is incredibly valuable. The authors present an approach for estimating various source properties derived from the classic MTT model, which in my view constitutes the main contribution of the paper. However, no attempt is made to actually evaluate this "source term model".

The inter-comparison study with RxCADRE data presented in the paper specifically excludes the source term model, focusing instead on the two previously-studied components of BUOYANT (plume rise and dispersion).

While such results are still valuable, they do not substitute for proper evaluation of the derivations presented in Section 2 and, in the current state, provide no supporting evidence for the main contribution of the paper.

Fortunately, RxCADRE dataset is incredibly detailed and can be used to extend the evaluation to include the source term model. My recommendation for the publication of this paper would, hence, be contingent on the authors demonstrating the results for the following:

Comparison of RxCADRE observations to:

BUOYANT model with **observations** as source inputs (this is essentially what's currently included in the paper), with more details included on methodology (as per comments below)

BUOYANT model with **old** source term parameterization (fixed parameters)

BUOYANT model with **new** source term model included

Operational version of BUOYANT (if different from above)

First, we agree with the reviewer that RxCADRE is a useful dataset. It has also been well documented.

According to the referee's advice, we have done a comparison of the model predictions, including the source term module, with the RxCADRE data and presented those together with the results obtained using the original model (using measured input data, excluding the source term module). We have included this comparison in the revised manuscript.

This comparison includes one major challenge. Most of the actual trunks of the trees were not burnt in the RxCADRE experiments, although part of the down-and-dead fine wood was burnt. However, the operational BUOYANT model assumes that the actual trunks of the trees were burnt. This is a reasonable assumption in an operational model, as it has been primarily developed for describing intensive major fires. Following the reviewer's line of thinking, we felt that this comparison is nevertheless useful to illustrate the potential use of the operational model in the case of less intensive forest fires, in which the fire does not burn a major fraction of the actual main tree structures, i.e., the trunks of the trees.

We have also done one sensitivity model run related to the model evaluation and included this in the revised manuscript. The sensitivity case analyzes the impact of the change of fire input data on the predictions (measured as average values for the first hour of the fire, instead of the whole duration of 3 hours). In our opinion, this addition is useful in understanding the uncertainties of such modelling.

Unfortunately, there are no suitable datasets in the publicly available literature for evaluating the source term module directly against experimental data, including burnt tree trunks.

The main aim of developing the source term module for the buoyant plume dispersion model was actually to make the use of the overall model a lot easier for the rescue personnel. The required

input data for the refined model is indeed substantially simpler and much easier to estimate, compared with those of the original model (i.e., the model excluding the source term module). However, this version of the source term module is not directly applicable to all conceivable forest fire cases. In major forest fires, which are the most important for rescue operations, also the tree trunks, or at least part of the tree trunks will be burnt. These arguments and model limitations are described more clearly in the revised manuscript.

Revisions made to the manuscript:
- An introductory section was added to the beginning of section 4, to make it clear, which model versions will be evaluated (including or excluding the source term module) and why; and for introducing also the related sensitivity model run.
- We have added an evaluation of the BUOYANT model, including the source term module. This is in section 4.4. Model evaluation. We have added a discussion regarding the results on including or excluding the source term module, and about the model sensitivity regarding input data (especially Fig. 5b and the related discussion).
- The whole section 4 has been re-checked and partly written more clearly and explicitly.
- Conclusions. We have improved and elaborated the discussion on the limitation of the source term module regarding including the burning of tree trunks.

Lastly, Section 4 of the paper is dedicated entirely to an overview of an operational modelling system. It is my understanding that the system is supposed to be accessible online, however no links are provided in the paper (aside from those pointing to an offline archive of the Fortran source code for the BUOYANT model). My current review of Section 4 is, hence, fairly superficial. If the authors are unable to provide access to the model for peer-evaluation, my recommendation would be to exclude this section from the manuscript.

We apologize for not specifying a link for the operational model version. The link is http://ilmanet.fi/order/147135, user name: GMD, password: FLARE21!. The FLARE user interface has currently been written only in Finnish. We have therefore provided the translations of the relevant texts to English as a separate document; this attached file is called 'Operational model - User interface - Translated to English'. In the following, there is some additional guidance for using the operational model.

Page 1: Login, user name: GMD, password: FLARE21!

Page 2: General information concerning the location of the burning and time of forest or pool fire. E.g., select 'Forest fire'.

Page 3: Editing of parameters describing weather and forest. Weather parameters are produced automatically by the NWP model HARMONIE.

Page 4: Same as 'Page 3', but the user can define weather parameters him/herself.

Page 5: 'Pool fire' option selected.

The details on how to download the BUOYANT code and the relevant data are described in the section 'Code and data availability'. These contain the source code of the BUOYANT model, the technical reference of the model, the user manual and the model input data corresponding to the work described in this paper.

**Specific Comments**

**Section 1**

The current overview is rather scattered, jumping from plume rise, to combustion, to air quality modeling , to emissions, to CFD models and, finally, to BUOYANT, with little effort to connect the topics. The broad context should be covered in a more concise and logical way (e.g. are satellite emissions studies and CFD models even relevant?), while the work pertaining directly to BUOYANT model needs to be covered in greater detail.

We have re-structured the introductory section. The underlying idea of this structure is to (after a general overview paragraph) first address the analysis of the fires as such, then emission modelling, subsequently dispersion modelling, and finally to set the present study into the context of the previous research.

Satellite measurements are important, especially for estimating the fire radiative powers of fires, which can then be used in chemical transport models. This is important especially if fires in regional or continental regions have to be analyzed. These also make it possible to analyze fires online. This research team has developed one of the leading models in this area (first described by Sofiev et al., 2009; online fire forecasts are available at https://silam.fmi.fi/). We have tried to describe this better in the revised introduction.

The CFD models have been used and are relevant for liquid pool fires; however, not (at least not yet) for wildland fires; this has been clarified in the revised manuscript. We refer here to the fairly long list of references in this part of the introduction, which have addressed such modelling.

The current version addresses, in this order:
- the categories of major fires and their overall impacts,
- nature and stages of burning in wildfires
- wild-land and liquid pool fire emission models
- dispersion modelling of fire plumes in the atmosphere
- overview of the previous research of this research team in the area of fire dispersion modelling, and the analysis of the input data for the model
- objectives of the study

In our view, the literature in this field is fairly scattered and disparate. There are specific studies on each of the above-mentioned topics, but not many reviews of the whole area.

*Line 34: Please include more recent review works on plume rise from wildfires*

We have included three more recent reviews (Lareau and Clements, 2017; Paugam et al., 2016; Val Martin et al., 2012). These are in the introductory section, the 7th paragraph.

*Line 35: How does this section connect to the previous one? How are combustion studies relevant?*

We have restructured the introduction, and this discussion has been therefore moved to a more logical place later in the introduction.

*Line 61: "satellite-based estimates of wildland fires": what does that mean? Emissions from wildland fires?*

We have revised this to read 'satellite-based emission estimates of wildland fires'. This is more accurate.

*Line 63, 65: why should measuring FRP "require inputs"? How are FRP-based emissions estimates relevant to this work?*

Satellites provide directly values of FRP in each satellite geographical pixel. This FRP value can be used for modelling the emissions of wildfires, without any additional information, e.g., the area of the fire, what material exactly is burning, etc. One such method has been described in detail by Sofiev et al., 2009, and the resulting online fire forecasts are available at https://silam.fmi.fi/.

This is relevant for the current study, as it is a complementary method.

We have added the following new paragraph to the introduction:
The satellite-based methods can be considered to be complementary in scope, compared to the detailed emission, plume rise and dispersion modelling presented in the current article. The main advantages of the satellite-based methods are that these can be used online and for extensive areas containing wild-land fires. Limitations of the satellite-based methods include that these cannot detect smaller fires; these also do not contain any direct information on the nature of the fire (e.g., forest fire or industrial fire). If more detailed and accurate analyses for any specific fire will be needed, then it is advisable to use a dedicated fire emission and plume rise model.

*Line 71: how does this section connect to the previous one and what is the relevance?*

This paragraph addresses liquid pool fires. This is one of the two main types of fires considered in this study.

*Line 80: Given the overview of literature provided, what are the current knowledge gaps this manuscript is going to address? What is missing?*

The most important matter that has been missing operationally has been a more straightforward analysis and provision of the input data for the plume rise modules. The required input data is complex, and not at all easy to assess. In previous literature, no modules have been presented for the source term evolution of major fires.

The uncertainties of the input data concern both the met data and the required data on the fires. Regarding the latter, e.g., an accurate and reliable analysis of the mass and heat fluxes from a fire is far from easy.

We have revised the paragraph explaining this in the introduction, the current version reads: The input data for the BUOYANT model has included various meteorological parameters and information on the properties of the fire and its surroundings (Kukkonen et al., 2014). However, a reliable determination of the required input data is a challenging task. This has up to date been the case also for all other models for the dispersion of buoyant plumes from major fires. The application of such models has therefore been possible only for expert users. In previous literature, no models have been presented for the source term evaluation of major fires. This has been a major obstacle to a more robust and widespread use of such plume rise models.

The physical and chemical limitations of the BUOYANT model have been analyzed in more detail in the methods section.

We have included two new paragraphs that discuss the model limitations, in section 2.1 'Overview of the modelling system', as follows.

The limitations of the BUOYANT model include: (i) The model assumes a steady state in terms of emissions and meteorology. However, the user can conduct multiple runs with various values of the emissions and meteorological parameters, to analyze the impacts of changing emissions and ambient conditions. (ii) The current model version does not treat the impacts of phase changes of water in the plume (in particular, condensation and evaporation). (iii) The model adopts some values of model parameters according to the best available previous experimental and modelling studies. (iv) The chemical reactions of pollutants during the source term and plume rise stages are not addressed.

The source term module has some additional limitations: (i) the source term module predicts the amount of the burning material based on the properties of the living trees within the burnt area, but it does not explicitly include the pollution originating from burning litter, down-and-dead fine wood, shrub and herbaceous plants, (ii) the emissions of pollutants originating from the burning of

wood are based on semi-empirical coefficients, determined in specific conditions, which may not be representative for all conceivable forest fires, (iii) the intensity of burning in forest fires depends also on the climatic conditions, previous and current weather, the species of trees and other plants within the burning area, and the amount of water in the burning material; these factors are not explicitly addressed by the source term module.

*Line 81-87: please include more detail about literature on the BUOYANT model and the relevant findings*

This is a reasonable request. We have added a more comprehensive description of the model structure and the main findings. For additional details, the reviewer is referred to our previous article on the BUOYANT model in GMD, Kukkonen et al. (2014). We have also extended the discussion on the comparison of the model against the wind tunnel measurements.

*Line 93: what do authors mean by "all other models"?*

We have added references to two recent studies that address such models.

*Section 2*

*Figure 1(b). Does the model actually include full vertical profiles, or is it a two-layer atmosphere? I.e. please move all information from Appendix D into the body of the paper. It's critical for understanding how the model works.*

BUOYANT assumes a three-layered atmospheric structure, allowing for full vertical profiles. We have moved Appendix D to the methods section, this is section 2.2. in the revised manuscript.

*Line 139 - 150: This belongs in Introduction*

Yes, we agree. We have moved this paragraph into the introduction.

*Line 148: Which "model"? What were the specific findings? If the agreement was great, why would BUOYANT need improvement? What were the limitations?*

Previous validations have been performed for the plume rise and atmospheric dispersion modules of the BUOYANT model (but not including the source term module). These results have agreed with the observations fairly well. The major limitation has been the analysis of input data for the model. The current manuscript attempts to bridge that gap.

This discussion has been clarified in the revised text.

The main aim of developing the source term module for the buoyant plume dispersion model was practical, to make the use of the overall model easier for the rescue personnel.

*Line 151: What do authors mean by "treatments"? Have the above-mentioned model evaluation studies all relied on these three fixed parameters? What are the parameters?*

With 'treatments' we referred to the physico-mathematical modelling. We have revised this sentence. The parameters are the along and cross-plume air entrainment coefficients, and the so-called added mass term (which accounts for the plume having to push air out of the way as it rises). All the model evaluation studies, including the current RxCADRE study, have relied on the same values of the three parameters (cf. lines 153-154.), as determined previously against the wind tunnel experiments. This paragraph has been revised to be more explicit (listing the parameters) and more clear.

*Line 174: Which "separate models" are the authors referring to?*

We have added references to survey articles (Sullivan, 2009a-c) of models of fire spread (propagation of fire in terrain).

*Line 175: I struggle with this assumption for wildland fires. Winds modify convective fluxes, also wildfires generate their own winds. Please provide more support for this. A thorough model evaluation with observational data would, of course, constitute the best supporting evidence.*

Correct, there may be significant "two-way interactions" between the ambient flow and a buoyant fire plume. However, this chapter and specifically the line/paragraph describes only the source term modules for forest and pool fires (not the whole BUOYANT model). The source term modules are applied only in the vicinity of the ground (up to the mean flame height, ~ 1-10 m).

We do recognise that ignoring the interaction *after* the source region would not be warranted. However, the plume rise module in BUOYANT allows for this, such as e.g. the fire plume to bend over by the ambient wind.

We have clarified this paragraph.

*Line 200: How does the model handle fuel moisture then? It's a critical parameter for wildland fires.*

BUOYANT assumes both the atmosphere and the fire plume are dry. This is a shortcoming of the model, which may, for instance, result in an underprediction of final plume rise due to missing latent heat contribution. This is a topic of further model development.

*Line 224: So which terms are going to be used in the subsequent modelling steps? A summary table would be helpful.*

We agree about this comment. A summary of the applied equations of the source term module is presented in section 2.3.6. in the revised manuscript.

*Line 257: Please clarify further what atmospheric stability structure is considered. Does "calm" refer to neutral stably stratified vertical profile?*

The MTT model applies to unstratified, no wind atmospheric conditions. This concerns only the source term regime. This has been clarified in the revised manuscript.

*Line 258: MTT applies specifically to point sources*

Correct, the classical (original) MTT model applies to point sources. However, area sources are often modelled by introducing a virtual source concept (e.g. Morton et al., 1965; Fanneløp and Webber, 2003). The source term module of BUOYANT describes area sources with a virtual point source below the actual fire source (cf. Eq. (9)). Further, the source term Eqs (8a-c) of BUOYANT are experimentally adjusted forms of the MTT equations. A note on the applicability of the original (unmodified) MTT model to point sources (only) has been added to section 2.3.3.

*Line 280: The original entrainment assumption of the MTT model doesn't hold for wildland fires. Has the RS entrainment model been used for such scenarios? If so, please provide a reference. If not, please provide support for the assumption.*

Correct, the entrainment assumption of the original MTT model (Eqs. 6a-c, 7a) is not applicable for wildland fires. However, the original MTT model (with MTT- or RS-entrainment) is not utilized in the source term module of BUOYANT. Instead, the source term module is based on experimentally adjusted forms of the MTT equations (Eqs. 8a-c) more suitable for highly buoyant plumes. The modified forms are usually referred to as strongly buoyant or strong plume relations (Heskestad, 1984 and 1998).

Entrainment in the plume rise phase (after the source term) in BUOYANT is estimated with a Ricou-Spalding (RS) type entrainment model modified to account for the bending over of the plume as it rises in a windy atmosphere. The plume rise module can be described to be an extension of the original MTT-model accounting e.g. vertically varying atmosphere and the interaction of the plume with an atmospheric inversion layer. The (modified) RS entrainment model has been applied in the previous model evaluation studies of the BUOYANT model (Sofiev et al., 2012; Kukkonen et al., 2014).

*Line 316: How does this connect to the previous paragraph? As mentioned above, a summary table of modeled parameters can likely help with clarity and structure.*

This is a good point. Therefore, we have divided the section "2.3.3 Radius, velocity, temperature and molar flux of a fire plume" into two sections ("2.3.3 Radius, velocity and temperature of a fire plume", "2.3.4 Molar flux of a fire plume").

At the end of section 2.3. (last paragraph), we have summarized and explained the modelled parameters.

*Eqn 10: At which height? I.e. this is not considering entrainment/detrainment. Please clarify the assumptions used for your definitions*

The total molar flux ($q_n$) is modelled at the mean flame height ($\lambda$), which is assumed to be the elevation of the source (term) for the plume rise module. Entrainment is (implicitly) included since the radius ($r$), velocity ($u$) and temperature ($T$) are solved from the Eqs. (8a-c) and the model of Appendix B. Detrainment is not considered in any module of BUOYANT.

We have clarified the introduction to equation (10) to read: "Assuming ideal gas behaviour, the molar flux of gaseous material ($q_n$) through a circular source plane is …". Assuming the ideal gas law and the assumed geometry:

$$pV = nRT \iff n = \frac{pV}{RT} \Rightarrow \frac{dn}{dt} = \frac{p}{RT}\frac{dV}{dt} \iff \frac{dn}{dt} = \frac{p}{RT}\pi r^2 u,$$

which is equ (10).

*Line 348: Which "model" and which "computer model". Please see my suggestion in the Writing Style section.*

We have changed the title of section 2.2.5 (section 2.3.6. in the revised manuscript) from "Summary of the use of the model equations and the computer model" to "Summary of the use of the source term module equations and the computer implementation".

*Section 3*

*As noted above, my main concern with this model evaluation is that it ignores entirely what has been presented in the previous parts of the paper. Moreover, evaluation of the plume rise and dispersion components of BUOYANT has been previously performed, based on the authors' own Introduction.*

*What is then the main goal and novelty of this section then? It is critical that this portion of the manuscript is expanded to include a full evaluation of the source term module (see General Comments above).*

We have expanded this section. Results have also been presented including the source term module and for an alternative manner of analyzing the input data. These changes were explained in more detail as a response to the reviewer's general comments.

*Please make a table of all required model input parameters, showing the estimates based on (a) RxCADRE data and (b) the source term model presented. Please include a description of how each value was obtained, pointing to specific equations, data sources etc.*

We have added two tables (Tables 1 and 2) of all required model input parameters showing the estimates based on (a) RxCADRE data, including a brief description of how each value was obtained; also the data source is indicated. Also, when appropriate/applicable, a reference to a specific equation is given.

*Lastly, please include a detailed discussion (either under model evaluation, or as a separate section) addressing model performance and placing it in context of earlier published work.*

We have substantially revised and expanded the discussion on the modelled results.

*Line: 406: Please include the following information: dimensions of the burn, how it was inited (and how long was the ignition process), duration of the active burning.*

We have added information on the dimensions of the burn, how it was ignited and the duration of the ignition process (first paragraph of section 4.2 in the revised manuscript). The duration of so-called active burning (depending on how exactly that will be defined) can be seen based on Fig. 3 in the revised manuscript.

*Line 420: Which heat fluxes, specifically? How was FRP estimated?*

The heat fluxes are the radiative heat fluxes modelled using long-wave infrared (LWIR) measurements (Dickinson et al., 2016) (cf. line 406).

A description of the determination of FRP from the LWIR measurements has been added to the second paragraph of section 4.2 of the revised manuscript.

*Line 450: There are vertical atmospheric soundings that were collected during the experimental campaign (see Clements 2016). Are these used for the study? Or does it solely rely on 30m tower observations? If so, please explain why.*

Both the 30 m tower observations and lidar measurements were used for this study (cf. lines 443-447 of the discussion paper).

*Line 456: Please create a Figure showing the observed vs. modeled atmospheric profile.*

We have added Figs. presenting experimental and modelled vertical profiles of temperature, wind speed and wind direction, Figs. 4a-c.

*Line 458: Were the winds considered to be uniform? Is BUOYANT able to account for vertical wind shear? Please move all the answers to these questions out of Appendix D and into the paper.*

Vertically the winds were not considered to be uniform, see, for instance, the newly added Figs. of experimental and modelled atmospheric profiles (Fig. 4 of the revised manuscript). Laterally and temporally the winds are assumed by the BUOYANT model to be uniform. Vertical wind shear is taken into account.

Appendix D material has been moved to the main paper, section 2.2.

*Line 466: The paper presents a new source term model. Model evaluation, therefore, CANNOT exclude the source term parameterization.*

We have included results including the new source term module (Fig. 5b). We have also added one model run that assumes the input data measured during the high-intensity period of the fire. Both of these results have been discussed in the text.

*Line 471: "the BUOYANT model" as per Line 121, includes source term module. As noted above, please provide the estimates of all model input parameters using the source term module.*

The model input parameters have been described in much more detail in the revised results section.

*Line 484: Please plot the used FRP and area values on Figure 2 as a horizontal lines. They seem a bit low to represent the average (though my eye-balling could be wrong). It would be great, if the authors presented some sort of sensitivity analysis of the model to these input parameters.*

This is a good suggestion. Applied average FRP and area have been added to Fig. 3 of the revised manuscript as horizontal lines. We have done a sensitivity model run, using the input during the high-intensity stage of the fire.

*Line 486: Abstract states that the BUOYANT model estimates cross-plume integrated values. My understanding is that in the current section the authors are presenting instantaneous concentrations. Please explain the discrepancy.*

We have revised this sentence in the abstract to read more clearly as:
"The model addresses the variations of the cross-plume integrated properties (i.e., the average properties along a trajectory) of a rising plume in a vertically varying atmosphere, and the atmospheric dispersion after the plume rise regime." The model predicts the evolution of the average properties in each cross-section of the plume, and then applies an assumed functional distribution of values (e.g., concentrations) in any cross-section. Consequently, the model predicts also (instantaneous) concentrations as a function of space and time.
We hope that this clarifies the matter.

Both the observed and modelled concentrations in the RxCADRE comparisons represent 20 seconds centred moving averages (along the flight path of the measurement aircraft).

*Line 507: "temporally well-captured" is a bit generous. Figure 3(a) estimates plume rise to be at half the measured value.*

Yes, we agree. This statement has been re-written.

*Line 513: Are these middle points in 3(b) calculated using center of mass? (they probably should be).*

Middle points in the manuscript were determined visually. However, the middle points in the revised manuscript have been re-computed using the centres of mass. This resulted in slight changes in these values; the visually determined middle points were approximately 10 % higher compared to elevations estimated by the centre of mass. We have revised the text and Fig. 3b accordingly. Minor refinements were applied also for the vertical extents of the plume.

*Line 519: There's quite a bit of a disagreement, actually (for good reasons). 4(a) shows no smoke above 590m, while observed values extend to 1230m. There's likely directional wind shear present, as higher observed values are shifted in the cross-wind directions. As noted above, a plot of vertical vs modeled atmospheric profile is much needed.*

Correct, directional wind shear was present, which can be seen in Fig. 4c (vertical profile of observed wind direction) of the revised manuscript. Above 1 km, the average wind direction deviates from the measured direction of more than 15 degrees. Further, the shift in wind direction can be seen also in the observed distributions of horizontal concentration in Fig. 4a (6a in the revised manuscript); the observed distributions at 920 m and 1230 m are shifted to the left from the average wind direction. The modelled concentrations at these elevations were zero (not shown in Figs.). However, the lack of the model not accounting for directional wind shear does not explain the disagreement between observations and the model above 900 m (or so). The likely explanation is the application of fire properties averaged over the whole measurement period, which tends to underestimate the more intense fire during the first hour. This has been corrected in Fig. 3 of the revised manuscript.

*Figure 3(b) drastically underestimates plume width. Once again, there's likely a logical explanation for this. Namely, as far as I recall, the lot was ignited with multiple straight lines (strip head fire ignition) of roughly 1km length, and the model attempts to represent the same fire strips with a circle of 158m radius (based on the configuration file provided in supplementary material). One shouldn't expect this assumption to work well in the given scenario.*

The average area of burning, based on the measured burning areas vs. time, has been presented in Fig. 3; it is approximately 0.08 km$^2$. This value of the area has been used in the modelling as the active fire area. However, it is correct that the inhomogeneous and even scattered aerial distribution of the fire within the burned area (that has been presented in the RxCADRE data files) will result in a more diffuse plume (with a wider plume and lower peak concentration), compared with the modelling. Another reason for the less wide modelled concentrations is the selection of the spatial distribution in the plume rise regime. These considerations have been discussed in the revised manuscript.

*Figure 3(c) again, suggests lack of wind shear and strong concentration overestimation. Please include an objective description of model performance here, as well as in Discussion. Also, please explain why the observed values at 280m (green line) in 3(c) simply cut off. Lastly, there appears be no plume growth and widening at all: it remains 2km wide from 1.7km to 11km downwind. Please explain. This, in part, explains the overestimation of concentration values further downwind.*

First, we have assumed constant meteorology in the modelling. In reality, some changes may of course have happened. E.g., the applied average wind direction in the model was derived from the measurements before the ignition in the vicinity of the L2F burn. The locations of the observed concentration distributions to the right compared with the predictions (e.g., Fig. 6c of the revised manuscript) at all elevations suggest a turn of wind direction in the later stages of the burn.

The overprediction (Fig. 6c in the revised manuscript) may be due to the application of the averaged fire properties over the whole measurement prediction; the averages tend to overestimate the fire properties during the later stages.

"Also, please explain why ..... in 3(c) simply cut off". We apologise that Fig. (4c) in the old manuscript was in error, which has been corrected in the revised manuscript (that is Fig. 6c in the revised manuscript). Corresponding changes were made to Fig. 5d of the revised manuscript.

"Lastly, there appears to be no plume growth and...": The DD values in the panels of Figs. 6a-c are the distances of the aeroplane during the whole measurement period of that flight. However, the horizontal measurements, as shown in the curves in the panels, represent shorter periods than indicated in the DD values. Figs. 6 have been revised accordingly.

The model properly predicts a dispersion of the plume: e.g., from 3.0 km to 6.3 km the plume width expands from 300 m to 430 m. This can be seen from the computed data, although this may not be clearly visible based on Figs. 6, due to the representation of the figures.
We have substantially improved and expanded the discussion of Figs. 6 in the text.

*Line 528: Please be more specific: what are the differences? What are the challenges? Wasn't the source term model designed to address these challenges?*

We have substantially improved and expanded the discussion of Figs. 6 in the text, presenting also better the differences between modelling and experiments, and the main challenges of such modelling.

*Line 541: What meteorological inputs are referred to, specifically? Full vertical atmospheric profile was measured during the RxCADRE campaign, so the vertical structure did not need to be estimated. Again, please include a figure for the vertical profile used in BUOYANT vs observations.*

Due to the treatment of atmospheric profiles in BUOYANT, the measured profiles cannot be applied directly. Instead, temperature and wind speed profiles were fitted into the observed profiles (cf. Figs 4a-c in the revised manuscript). We have added these figures to the revised manuscript, to show explicitly the observed and modelled atmospheric vertical profiles.

*Line 545: Given observed values were available, this argument cannot be used to explain the discrepancies between modelled and measured concentrations.*

We refer to the reply above to the previous reviewer's comment.

***Section 4.***

*Please provide a link to the online operational version of the model (if it's indeed available in the paper somewhere and I missed it, my apologies! Please move the link the beginning of Section 4.1 for easy access).*

Please see the reply above on p. 3., including a link to access the operational model.

Unfortunately, due to legal reasons associated with our funding organizations, we can grant external organisations access to the web-based operational system only if requested. However, the source codes of the BUOYANT program, the technical reference of the model, the user manual of the model and the model input data corresponding to the work described in this paper are publicly available, as described in the section 'Code and data availability'.

*Without access to the system, the following comments do not constitute proper peer-review. Please also include the operational version of the model in the evaluation.*

Please see the previous response.

*Line 632: Ambient pressure, wind and temperature profiles are standard model output. What's being "evaluated"?*

Monin-Obukhov length required by the BUOYANT model is calculated from the HARMONIE products, sensible heat and surface momentum fluxes (as the Monin-Obukhov length is not directly in the output data of HARMONIE). The model assumes a two-layered thermal structure above ABL (inversion and upper layers), which is estimated by applying the HARMONIE predictions with a method modified from Fochesatto (2015). So, the profiles of HARMONIE have to be pre-processed to be applied as input to BUOYANT. The background reason is of course that HARMONIE has been developed for numerical weather forecasting, not for air quality studies.

*Line 635: Is roughness length the only parameter that actually needs estimation?*

Roughness length is obtained directly, without any processing, from the CORINE (CoORrdination of INformation on the Environment) land cover information (cf. lines 636-638). The incorrect wording "...is evaluated based on the CORINE..." on line 636 has been changed to "...is extracted from the CORINE...".

All other input information of BUOYANT related to meteorology requires preprocessing of the products of the HARMONIE weather prediction model.

*Line 660: What do authors mean by "efficient functioning"?*

This sentence was removed.

***Conclusions:***
*Here the authors seem to mix summary with discussion. Please see my earlier comment, regarding clearly separating out results from discussion.*

We have re-written the conclusions. First, we have added a better description of the underlying rationale of this study (two first paragraphs). Second, we added one paragraph that describes the main limitations of the model. Third, we wrote the conclusions on the comparisons of model predictions and data more specifically (based on the more detailed model runs and their interpretation in the revised manuscript). Fourth, we addressed specifically one key uncertainty related to the estimation of the amount of burnt material.

*Line 665: As above, please clarify whether this is a cross-plume integrated model or not. Figure 4 would only be possible if the fields are NOT cross-plume integrated.*

This has been revised.

*Line 671: which characteristics, specifically?*

We have written these explicitly in the revised manuscript.

*Line 684-687: Description of the campaign belongs in the intro/methods, not in conclusion.*

We have removed the description of the campaign from the conclusions.

*Line 687-691: Please see comments to Section 3.*

These statements have been rewritten.

*Lines 693-712: This very much resembles a Discussion section, and should be relocated away from Conclusions, accordingly, together with responses to comments for Line 519 (please see above)*

These statements have been removed from the conclusions.

*Line 696: What are the challenges, specifically? What are the major sources of uncertainty, specifically?*

We have added a paragraph on the main limitations of the model (the 4$^{th}$ paragraph of the conclusions).

***Appendix C***
*Line 935: that seems to be a fairly extreme assumption, I am very curious to see how the source term model performs for the L2F burn.*

Yes, this could be considered extreme. However, it has been detected to take place in major high-intensity fires. However, we agree that this is one key aspect of further model development (as we have stated in the revised conclusions).

***Appendix D***
*Please move this into the main body of the paper.*

This has been moved to the main text.

*Writing Style*

*The authors use the term "evaluate" extensively, to refer to both "estimation" and "measurement" of a particular parameter. My recommendation would be to specifically indicate whether a particular term is estimated (i.e. modelled) vs. measured (i.e. obtained from observations) vs. calculated.*

OK. We have checked and revised these throughout the manuscript; removing excessive "evaluate" wordings. However, we wish to use the conventional term 'model evaluation', when referring to the comparison of model predictions and measured data. In some cases, parameters are not directly measured, but instead analyzed or assessed based on measurements; we have tried to find the right wording also in such cases.

*The variable use of term "model" to refer to BUOYANT with multiple components, as well as to BUOYANT without the source term model, to the source model and to the computer model is rather confusing. My suggestion would be to call BUOYANT (with all three components) a model, and individual sub-models (dispersion, plume rise, source term) - modules. Operational version can be referred to as BUOYANTops, or something on those lines.*

We have changed the terminology as suggested. The individual sub-models of BUOYANT are called modules. The operational program has already been named FLARE (Fire pLume model for Atmospheric concentrations, plume Rise and Emissions).

*Please note that the draft requires extensive English language editing.*

We have checked and improved the English language throughout the whole manuscript.

**Authors' preliminary comments to Referee no 2**

General Comment:

The paper presents the development of a source term model that evaluates the fire plume properties just above the flame as an extension to the previously published BUOYANT model for the dispersion of buoyant plumes from wildfires and liquid pool fires under varying atmospheric conditions. The refined BUOYANT model v4.20 is then evaluated against observational data of $CO_2$ concentrations from aircraft measurements during a wildfire experiment from the RxCADRE campaign. The model captured well the vertical profiles of $CO_2$, while the highest concentrations were moderately overpredicted. The authors also state that the widths of the plumes are slightly underestimated, without giving an explanation for this behavior.

We have substantially expanded the discussion on the differences in model predictions and measured data in the revised manuscript, also including the widths of the plumes.

Further, an operational version of the BUOYANT model, called FLARE, is briefly presented, although it remains somewhat unclear how it is related to the research version of the model.

These model versions are closely related to each other. Both the operational version of the model (named FLARE) and the original research model (named BUOYANT) use the same code for the dispersion and transport of a buoyant plume. The operational model includes the whole of the research code, and all the core physics computations in the operational model are done using the research code. However, there is a dedicated user interface in the operational model, to facilitate easier use.

In more detail, the differences between FLARE and BUOYANT are:

(a) FLARE uses as default the presented source term module; it does not, therefore, allow the user to specify the source related input in a more complex format

(b) The specification of the meteorological conditions in FLARE is determined by the used numerical weather prediction model, in BUOYANT, this can be done also in various other ways.

(c) Output of BUOYANT is more versatile and can be adjusted by the user, compared to FLARE.

(d) BUOYANT allows the user to post-process the model results at will. FLARE includes standard format post-processing.

(e) FLARE can be used with a restricted set of web browsers, while BUOYANT is designed to be as platform-independent as possible.

These connections have been explained in more detail in the revised manuscript. In the first paragraph of section 2.2., the differences in terms of providing the meteorological data have been described. The other differences between these model versions have been described in section 3.1, the last paragraph.

Currently, the evaluation against the experimental data from the wildfire plume is performed with the refined BUOYANT model, but not including the extension with the new source term model. It is a bit unfortunate that the validity of this essential new module has not been demonstrated. My recommendation is to carry out a dedicated sensitivity study of the possible input value ranges in the source term model and after interfacing it with the BUOYANT model, comparing to the experimental data.

We have included also a comparison of the source term module predictions with the RxCADRE data in the revised manuscript. Although the amount of wood burnt cannot be accurately predicted with the source term module, in that case, we felt that this comparison is useful to illustrate the potential use of the operational model in the case of less intensive fires. According to the reviewer's suggestion, we have also done one sensitivity model run, regarding the impact of the measured fire intensity on the predictions.

Section 4.4. has been revised accordingly.

Nevertheless, I think that the development of a physical model that can deal with the early evolution of the fire plume, the plume rise, and the local atmospheric dispersion is of great value for achieving advances in the prediction of impacts from major natural and man-made fires. Overall, the paper deserves publication after my general comment and the specific comments below are sufficiently addressed.

1.) P. 2, lines 48-51: It is mentioned that hundreds of chemical compounds are emitted into the atmosphere during wildland fires. Table C1 also lists several different fuels for which the convective heat flux and mass fluxes during a liquid pool fire can be calculated. How much of the combustion chemistry and oxidation is BOUYANT capable of simulating?

The mass burning rates (i.e., rate of mass burned per time) from a liquid pool or forest fire are estimated with semi-empirical correlations (these have been described in detail in Appendix B of the revised manuscript). Estimates of substance-specific fire products (e.g., the mass flux of $CO_2$) are obtained from the modelled mass burning rate and experimentally derived emission factors. Currently applied substance-specific emission factors for liquid pool fires are shown in Table B1 of the manuscript. For forest fires, we have applied the substance-specific emission factors presented by Kaiser et al. (2012).

However, after the emissions have been modelled for a range of chemical compounds, the model does not address chemical reactions within the source term and the plume rise regimes. However, the model can predict the plume properties after those regimes and then be coupled with dispersion models that address chemistry. For instance, the BUOYANT model has previously been used in that way in combination with the chemical transport model SILAM. The SILAM model contains a chemistry sub-model including more than a hundred chemical compounds and their chemical reactions in the atmosphere. In this way, the model can be used as part of a larger modelling system that includes also atmospheric chemistry.

As far as we know, none of the publicly available plume rise models or modules treats explicitly the chemical reactions during the plume rise regime.

2.) I suggest to revise the paragraph on CFD models in the Introduction (P. 3, lines 70-80) to address the treatment of plumes from the two different types of fires: wildfires and liquid pool fires. Currently, only the dispersion models for treating liquid pool fires are described. Further, the respective description should deal with above-fire (source term), plume rise and large-scale dispersion for the two fire types.

The CFD models have been used for estimating liquid pools fires, as described in the manuscript. However, we are not aware of studies, in which these would have been used for analyzing forest fires. We have revised this paragraph to be clearer. We also included a specific sentence referring to wild-land fires.

The physics of plume rise modules, such as the plume rise module of BUOYANT used in this study, is the same irrespective of the source of the buoyant plume. That is the reason for using only one computer module for analyzing the plume rise regime (after the source term regime), also in the BUOYANT model. The source term module has to be different for the forest fire and pool fire cases.

No previous modules have been presented for the source term evolution of forest or pool fires. That is why we have not discussed such modules or models. However, we have written this more clearly in the revised manuscript.

At which point, i.e. distance from source and vertical layer, does a large scale atmospheric model take over?

The criteria for the termination of the plume rise stage have been described in detail in our previous article, Kukkonen et al. (2014) (section "2.1.6 Criterion for the termination of plume rise"). The criterion used in the BUOYANT model is the following: "In the current model version, we have chosen simply to use the height, at which buoyancy force first vanishes as the final rise height of the plume." In short, when there is no buoyancy left in the plume, the model makes the transition from the plume rise regime to the dispersion modelling regime.

This distance (and the corresponding height) depends a lot on the intensity and area of the fire, and the meteorological conditions. In the RxCADRE model simulation presented in the article the plume rise stage comes to an end at the distance of 6.2 km from the source, at a height of 510 m (the height at the plume centerline).

We felt that this is important information regarding the simulations and therefore added one paragraph explaining when the atmospheric dispersion model takes over (section 4.4, second paragraph).

3.) Section 2.2: it is not really clear from the descriptions in this section, how the new source term model for evaluating the properties of the fire plume above the flame tips is interfaced with the plume dispersion in the BUOYANT model.

This is a good point. This transition has been described in section 2.2.4 (from the source term to the plume rise regime). However, the description is probably too brief and general. We will add a more specific description to that section. This has been described in our previous article Kukkonen et al. (2014), quote: "Information on the source term (to be input to the plume rise module) includes the following: the source radius, the source height above the ground, the temperature of the released mixture of contaminant gas and particles, and air, the mass flux of this mixture, the mass fraction of the released gas, and the molecular weight and heat capacity of the released gas.".

We have added a description of these properties to section 2.1 (fourth paragraph in the revised manuscript).

4.) Section 3.3: as it is now, the evaluation is done for the BOUYANT model without the source term model. This may appear reasonable at first since the fire properties for the selected wildfire case L2F are well characterized by the observations. However, in order to evaluate the source term model presented in this paper, it would make sense to conduct a sensitivity analysis, studying the probable value ranges of selected fire parameters, and comparing the outputs of the BOUYANT model - including the source term sub-model - against the measurements in L2F.

We have included a sensitivity run to the revised manuscript regarding the intensity of the fire. The predictions of the source term module have also been included.

5.) Section 3.3.1: which method was used to evaluate the meteorological parameters on P. 18, lines 458-464, for use in the BOUYANT model?

The methods to model the Monin-Obukhov length, and the vertical profiles of wind speed and temperature have been substantially better described in the revised manuscript in section 4.3.1.

Atmospheric pressure was determined utilizing the measured ambient temperature profile and the hydrostatic assumption. The measured vertical temperature profile indicates (by visual inspection) a boundary layer height of approximately 2.2 km. The prevailing wind direction was assumed to be the arithmetic mean of the experimental wind directions at the heights 6.2 m and 990 m.

6.) Section 3.3.2: a table should be provided that contains the required input parameters and the values used in the evaluation both for the BUOYANT model without the source term model (currently listed on P. 19, lines 483-484) and for the BOUYANT model when using the source term model.

This is a good idea. We have compiled tables containing the input parameters of the model, as used with and without the source term module. These are in sections 4.3.1 and 4.3.2 of the revised manuscript (especially Tables 1-3).

7.) P. 23, line 536: the temporal evaluation should also be shown in a plot, for example at 450 m height above ground at the plume centerline.

This is a reasonable suggestion. However, we have revised the sentence on line 535 in the discussions version of the manuscript: "However, the temporal agreement of the measured and modelled highest concentrations was good.". The aircraft measurements represent centred moving averages over 20 seconds, at certain average three-dimensional locations. The exactly corresponding spatial and temporal averages were computed with the model and compared with the measured values.

Although both of these have been carefully performed and archived, it is in our view an overstatement to state something about the temporal evolution of measurements vs. data (as both the time and location are simultaneously changing in aircraft measurements). This was also the opinion of reviewer number 1. However, we have compared the measured data and predictions for certain flight manoeuvres, such as the crosswind distributions in Fig. 4.).

8.) Section 4: I think the presentation of the operational version of the model should be placed before the evaluation chapter 3 to make it more visible to the readers.

Yes, this is a good suggestion also in our view. We have moved the operational section before the model evaluation section.

The operational version FLARE needs to be better related to the research version of the model. Does it use the new source term module?

Yes, we agree. We have described these relations in much more detail in the revised manuscript, as discussed above in response to the general comments of reviewer number 2.

Information should be added about the stakeholder groups that are targeted as potential users. It would be nice to include a screenshot from a real-world example application.

The stakeholders and users of the operational model currently include a wide range of emergency response personnel in Finland, the operational meteorologists at the Finnish Meteorological Institute (FMI), experts at the Ministry of the Interior in Finland and researchers at the FMI. We hope to expand the user group in the future to rescue personnel and experts in other countries.

We have added a sentence about the stakeholders and users into the first paragraph of section 3. We have also added a screenshot of a real-world application to Appendix D.

Yes, these will be added to the manuscript.

9.) P.28, lines 689-690: statement "for most of the highest concentrations" seems to contradict with the finding of moderate overprediction of the highest concentrations.

We have revised this statement.

Technical Corrections
Figure 4, middle plot: the green curves from modelling and measurements do not show well.

We have revised Figure 4b.

**References**

Kaiser, J.W., Heil, A., Andreae, M.O., Benedetti, A., Chubarova, N., Jones, L., Morcrette, J.-J., Razinger, M., Schultz, M.G., Suttie, M., and van der Werf, G.R.: Biomass burning emissions estimated with a global fire assimilation system based on observed fire radiative power, Biogeosciences, 9, 527-554, doi:https://doi.org/10.5194/bg-9-527-2012, 2012.

Kukkonen, J., Nikmo, J., Sofiev, M., Riikonen, K., Petäjä, T., Virkkula, A., Levula, J., Schobesberger, S., and Webber, D.M.: Applicability of an integrated plume rise model for the dispersion from wild-land fires, Geosci. Model Dev., 7, 2663-2681, doi:https://doi.org/10.5194/gmd-7-2663-2014, 2014.

Sullivan, A.L.; Wildland surface fire spread modelling. 1990-2007, 1: Physical and quasi-physical models, Int. J. Wildland Fire, 18, 349-368, doi:https://doi.org/10.1071/WF06143, 2009a.

Sullivan, A.L.; Wildland surface fire spread modelling. 1990-2007, 2: Empirical and quasi-empirical models, Int. J. Wildland Fire, 18, 369-386, doi:https://doi.org/10.1071/WF06142, 2009b.

Sullivan, A.L.; Wildland surface fire spread modelling. 1990-2007, 3: Simulation and mathematical analogue models, Int. J. Wildland Fire, 18, 387-403, doi:https://doi.org/10.1071/WF06144, 2009c.